# Mechanical activation of TWIK-related potassium channel by nanoscopic movement and rapid second messenger signaling

E Nicholas Petersen[1,2], Mahmud Arif Pavel[1], Samuel S Hansen[1], Manasa Gudheti[3], Hao Wang[1,2], Zixuan Yuan[1,2], Keith R Murphy[4,5], William Ja[4,5], Heather A Ferris[3], Erik Jorgensen[6], Scott B Hansen[1]*

[1]Departments of Molecular Medicine, The Scripps Research Institute, Scripps, Jupiter, United States; [2]Scripps Research Skaggs Graduate School of Chemical and Biological Science, The Scripps Research Institute, Scripps,, Jupiter, United States; [3]Division of Endocrinology and Metabolism, Center for Brain Immunology and Glia, Department of Neuroscience, University of Virginia, Charlottesville, United States; [4]Department of Neuroscience, The Scripps Research Institute, Scripps, Jupiter, United States; [5]Center on Aging,The Scripps Research Institute, Scripps, Jupiter, United States; [6]Department of Biology, Howard Hughes Medical Institute, University of Utah, Salt Lake City, United States

*For correspondence:
shansen@scripps.edu

**Abstract** Rapid conversion of force into a biological signal enables living cells to respond to mechanical forces in their environment. The force is believed to initially affect the plasma membrane and then alter the behavior of membrane proteins. Phospholipase D2 (PLD2) is a mechanosensitive enzyme that is regulated by a structured membrane-lipid site comprised of cholesterol and saturated ganglioside (GM1). Here we show stretch activation of TWIK-related K$^+$ channel (TREK-1) is mechanically evoked by PLD2 and spatial patterning involving ordered GM1 and 4,5-bisphosphate (PIP$_2$) clusters in mammalian cells. First, mechanical force deforms the ordered lipids, which disrupts the interaction of PLD2 with the GM1 lipids and allows a complex of TREK-1 and PLD2 to associate with PIP$_2$ clusters. The association with PIP$_2$ activates the enzyme, which produces the second messenger phosphatidic acid (PA) that gates the channel. Co-expression of catalytically inactive PLD2 inhibits TREK-1 stretch currents in a biological membrane. Cellular uptake of cholesterol inhibits TREK-1 currents in culture and depletion of cholesterol from astrocytes releases TREK-1 from GM1 lipids in mouse brain. Depletion of the PLD2 ortholog in flies results in hypersensitivity to mechanical force. We conclude PLD2 mechanosensitivity combines with TREK-1 ion permeability to elicit a mechanically evoked response.

## eLife assessment

This **important** study poses a provocative mechanism of channel activation of the mechanically activated ion channel TREK-1. The data provide **solid** evidence that the application of shear to cells causes a redistribution of both TREK-1 and an associated enzyme, PhospholipaseD2 in the membrane that increases the enzyme activity. The work offers a new mechanism, but note that this is only one possible method of channel activation, and mechanisms independent of PLD2 are also probable.

**eLife digest** "Ouch!": you have just stabbed your little toe on the sharp corner of a coffee table. That painful sensation stems from nerve cells converting information about external forces into electric signals the brain can interpret. Increasingly, new evidence is suggesting that this process may be starting at fat-based structures within the membrane of these cells.

The cell membrane is formed of two interconnected, flexible sheets of lipids in which embedded structures or molecules are free to move. This organisation allows the membrane to physically respond to external forces and, in turn, to set in motion chains of molecular events that help fine-tune how cells relay such information to the brain.

For instance, an enzyme known as PLD2 is bound to lipid rafts – precisely arranged, rigid fatty 'clumps' in the membrane that are partly formed of cholesterol. PLD2 has also been shown to physically interact with and then activate the ion channel TREK-1, a membrane-based protein that helps to prevent nerve cells from relaying pain signals. However, the exact mechanism underpinning these interactions is difficult to study due to the nature and size of the molecules involved.

To address this question, Petersen et al. combined a technology called super-resolution imaging with a new approach that allowed them to observe how membrane lipids respond to pressure and fluid shear. The experiments showed that mechanical forces disrupt the careful arrangement of lipid rafts, causing PLD2 and TREK-1 to be released. They can then move through the surrounding membrane where they reach a switch that turns on TREK-1. Further work revealed that the levels of cholesterol available to mouse cells directly influenced how the clumps could form and bind to PLD2, and in turn, dialled up and down the protective signal mediated by TREK-1.

Overall, the study by Petersen et al. shows that the membrane of nerve cells can contain cholesterol-based 'fat sensors' that help to detect external forces and participate in pain regulation. By dissecting these processes, it may be possible to better understand and treat conditions such as diabetes and lupus, which are associated with both pain sensitivity and elevated levels of cholesterol in tissues.

## Introduction

All cells respond to mechanical force, a phenomenon known as mechanosensation (*Hahn and Schwartz, 2009*; *Julius and Basbaum, 2001*; *Ranade et al., 2015*). Mechanosensation requires the dual processes of sensing and transducing mechanical force into signals that cells can interpret and respond to. Ion channels serve as one type of transducer. While certain channels have been demonstrated to directly respond to mechanical force in purified lipid environments (*Brohawn, 2015*; *Cox et al., 2019*; *Teng et al., 2015*), others, such as TRP channels, can be downstream components of multi-step signaling cascades (*Kwon et al., 2023*; *Wilde et al., 2022*) in cellular membranes, including those involving mechanosensitive G-protein-coupled receptors (GPCRs) (*Gudi et al., 1998*; *Lin et al., 2022*; *Storch et al., 2012*; *Wei et al., 2018*; *Xu et al., 2018*),.

One of these channels is TWIK-related $K^+$ channel subtype 1 (TREK-1), a mechanosensitive member of the two-pore-domain potassium (K2P) family known for its analgesic properties. TREK-1 plays a role in inhibiting neuronal firing and reducing pain through the release of potassium ions (*Honoré, 2007*). Intriguingly, the enzyme phospholipase D2 (PLD2) directly interacts with the C-terminus of TREK-1, activating the channel by locally producing phosphatidic acid (PA) (*Comoglio et al., 2014*).

Despite lacking a transmembrane domain, PLD2 exhibits mechanosensitivity (*Pavel et al., 2020*; *Petersen et al., 2016*). This property arises from mechanically induced changes in spatial organization of the plasma membrane. Palmitoylations of cysteine residues near its pleckstrin homology (PH) domain enable PLD2 to bind to a specific site composed of monosialotetrahexosylganglioside (GM1) lipids and cholesterol in the liquid ordered ($L_o$) region of the membrane (*McDermott et al., 2004*; *Yuan and Hansen, 2023*; *Figure 1—figure supplement 1A and B*). Anesthetics compete with palmitates at this site which we call an anesthetic-palmitate (AP) site (*Pavel et al., 2020*). Application of mechanical force disrupts the AP site, releasing the palmitates of PLD2 and allowing the PH domain to interact with phosphatidylinositol 4,5-bisphosphate ($PIP_2$) clusters in the liquid disordered ($L_d$) region of the membrane (*Petersen et al., 2016*). This, in turn, triggers enzyme activation through spatial redistribution and substrate presentation (*Petersen et al., 2020*; *Figure 1—figure supplement 1C*).

Notably, the C-terminus responsible for binding PLD2 is essential for TREK-1 mechanosensitivity in cellular membranes (*Brohawn, 2015*; *Chemin et al., 2007a*; *Patel et al., 1998*). Our previous studies showed the same C-terminus is essential for binding PLD2 during anesthetic activation of TREK-1. In that system, the anesthetic-induced release of PLD2 from GM1 lipids was responsible for all the TREK-1 current evoked by anesthetic in HEK293T cells (*Pavel et al., 2020*). These results prompted us to investigate whether PLD2's mechanical activation in HEK293T cells contributes to a mechanically evoked TREK-1 current. Analogous to our study on TREK-1 anesthetic sensitivity, here we show that the catalytic activity of PLD2 is indispensable for the generation of mechanically evoked TREK-1 currents in cultured HEK293T cellular membranes. We further show these PLD2-dependent TREK-1 currents result from the spatial patterning of the channel and enzyme within lipid nanodomains in HEK, N2a, C2C12, and primary neurons. In neurons, this regulatory process is influenced by astrocyte-derived cholesterol.

## Results

### Dependence on PLD2 for TREK-1 activation

To assess the involvement of PLD2 mechanosensitivity to a mechanically evoked current from TREK-1 channels, we conducted pressure current measurements in HEK293T cells both with and without the expression of a catalytically inactive K758R PLD2 mutant (xPLD2) (*Toschi et al., 2009*). We selected HEK293T cells for this study due to their minimal endogenous potassium currents, which allowed us to attribute the recorded currents specifically to TREK-1. The expression of TREK-1 was tracked using an EGFP tag attached to the channels C-terminus, and we confirmed successful expression of all constructs, observing their presence on the cell surface (*Figure 2—figure supplement 1A and B*).

In intact HEK293T cells, the overexpression of xPLD2 substantially inhibited nearly all mechanically evoked TREK-1 currents (*Figure 1A–C*). When currents were measured in the inside out configuration under negative pressure conditions (from 0 to –60 mmHg), they exhibited a reduction of over 80% from $0.181 \pm 0.04$–$0.035 \pm 0.013$ pA/$\mu$m$^2$, despite an overall increase in the total levels of TREK-1 (*Figure 2—figure supplement 1B and C*). This observation suggests that if the effect is through a specific PLD2-TREK-1 interaction, then removal of the PLD2 binding domain should also inhibit PLD2-evoked currents.

To investigate further, we eliminated the putative PLD2 binding site on TREK-1 by truncating the C-terminus at residue 322 (TREK trunc) (see *Figure 1—figure supplement 2A*). The pressure-induced current in TREK trunc decreased by more than 85%, measuring $0.025 \pm 0.013$ pA/$\mu$m$^2$ (*Figure 1C*, *Figure 1—figure supplement 2B*). Remarkably, this reduction was nearly identical to the results obtained when full-length TREK-1 (TREK FL) was expressed in conjunction with xPLD2. Importantly, previous studies have confirmed the functionality and mechanosensitivity of this similar truncated TREK-1 when reconstituted into crude soy PC lipids (*Brohawn et al., 2014a*).

Conversely, the overexpression of TREK-1 with wild-type (WT) mouse PLD2 (mPLD2) resulted in a substantial augmentation of TREK-1 pressure-dependent currents within HEK293T cells (*Figure 1A*, see *Figure 1—figure supplement 2B* for raw traces). The TREK-1-evoked currents were markedly significant, both with and without overexpression of mPLD2, when compared to xPLD2 (p<0.002 and 0.007, respectively; *Figure 1C*). HEK293T cells, like all cells, have endogenous PLD2 (enPLD2). The xPLD2 expression is >10× the endogenous PLD2 (enPLD2) that we previously showed outcompetes the binding of enPLD2 to TREK-1 under similar conditions (*Pavel et al., 2020*).

The threshold for PLD2-dependent TREK-1 activation proved to be quite sensitive. TREK-1 exhibited responsiveness to negative pressure (ranging from 0 to –60 mmHg, in a non-saturating regime) with a half maximal pressure requirement of approximately 32 mmHg. This pressure led to the generation of up to 200 pA of TREK-1 current (*Figure 1A*, *Figure 1—figure supplement 2C*), aligning with findings from prior investigations (*Brohawn et al., 2014b*; *Patel et al., 1998*).

As previously mentioned, previous studies demonstrated TREK-1 direct sensitivity to mechanical force in purified lipids, independent of PLD2 (*Berrier et al., 2013*; *Brohawn et al., 2014a*). To delineate a PLD2-independent (i.e., direct) component of TREK-1 mechanotransduction within a cellular membrane, we conducted a comparative analysis between cells expressing TREK trunc and mock transfected HEK293T cells (lacking TREK-1 expression). Our results demonstrated that TREK trunc exhibited a modest increase in current, measuring $0.046 \pm 0.023$ pA/$\mu$m$^2$ (p=0.08, *Figure 1C* inset),

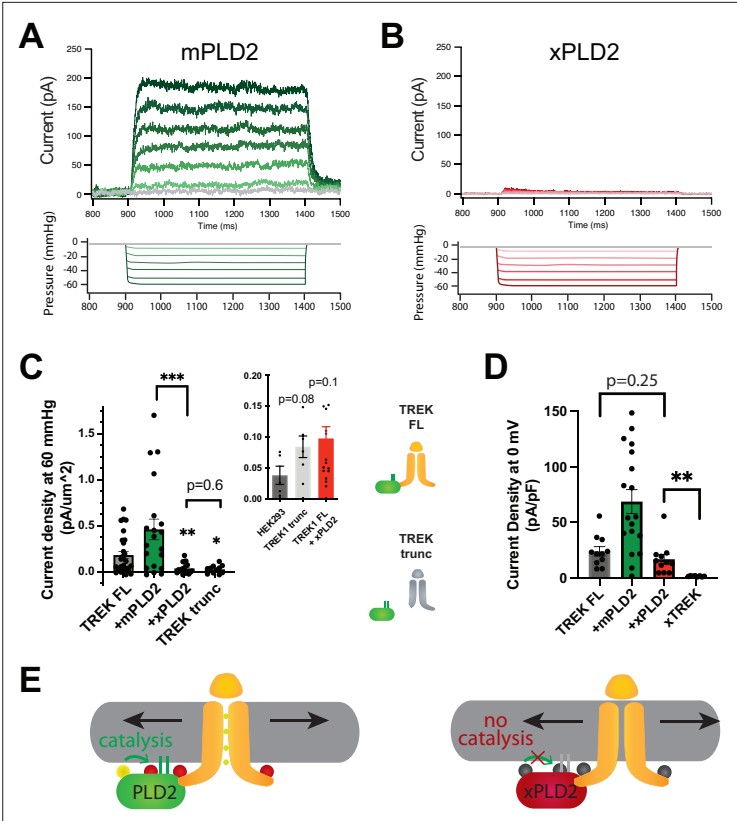

**Figure 1.** PLD2-dependent and independent mechanical activation of TREK-1 channels. (**A, B**) Representative traces from pulled patches of human TREK-1 overexpressed in HEK293T cells with mouse phospholipase D2 (mPLD2, green traces) (**A**) or catalytically inactive mouse PLD2 (xPLD2, red) (**B**) under pressure clamp (0–60 mmHg at +30 mV). (**C**) The data, after subtracting HEK293T background current (0.04 ± 0.02 pA/μm² n = 5 [inset]), are summarized for –60 mmHg. Compared to endogenous PLD2, the expression of xPLD2 eliminated the majority of detectable TREK-1 pressure current (p<0.007, n = 16–23), as did a functional truncated TREK-1 (TREK trunc) lacking the PLD2 binding site (p=0.002, n = 15–23). The inset compares mock-transfected HEK293T cells with TREK trunc and full-length TREK-1 (TREK FL)+xPLD2, indicative of direct TREK-1 activation. Asterisks indicate significance relative to TREK FL, except where noted by a bar. (**D**) Whole-cell TREK-1 potassium currents with and without xPLD2. TREK-1 is expressed and functional in the presence of xPLD2. A nonfunctional C-terminal truncation (C321) of TREK-1 (xTREK) is shown with no appreciable current HEK293T cells. (**E**) Cartoon illustrating PLD2-dependent TREK-1 opening in HEK293T cellular membrane. On the left, membrane stretch (black arrows) mechanically activates PLD2. When PLD2 is active, it makes phosphatidic acid (PA), which evokes the open state of TREK-1. On the right, in the absence of mechanically generated PA, the closed channels remain closed despite the presence of membrane tension. Statistical comparisons were made with an unpaired Student's *t*-test.

The online version of this article includes the following figure supplement(s) for figure 1:

**Figure supplement 1.** The role of lipids and lipid order in mechanotransduction.

**Figure supplement 2.** Electrophysiology details and methods.

when compared to the condition with no TREK-1. Similarly, in cells expressing xPLD2 alongside full-length TREK-1, the channel exhibited a small increase in current measuring 0.060 ± 0.035 pA/μm² (p=0.11). This PLD2 independent current aligns with direct mechanical activation of TREK-1 observed in liposomes. However, it is important to note that we cannot rule out the possibility of other contributing mechanisms.

Due to the minimal pressure-activated TREK-1 current observed in the presence of xPLD2, we conducted control experiments to validate the presence of functional TREK-1 on the plasma membrane—channels capable of conducting current. TREK-1 is a potassium leak channel with inherent basal currents in cultured cells (*Comoglio et al., 2014*). It is reasonable to assume that the basal current arises from the presence of basal levels of anionic lipids in the cellular membrane.

As anticipated, we observed TREK-1 basal current both in the presence (16.9 ± 4.3 pA/pF) and absence (24.0 ± 4.2 pA/pF) of xPLD2 (*Comoglio et al., 2014*; *Figure 1D*). These basal resting currents were recorded at 0 mV in HEK293T cells overexpressing TREK-1 FL with and without xPLD2. Notably, the current in the presence of xPLD2 was significantly higher than that observed in a nonfunctional control TREK-1 (xTREK), which we previously determined to exhibit no measurable current in unrelated studies. These controls affirm that TREK-1 FL is expressed and functionally active in the presence of xPLD2. Moreover, they highlight the essential role of PLD2 in pressure-activated currents (*Figure 1C*), contrasting with the basal leak currents (*Figure 1D*; TREK FL, gray bar vs. +xPLD2, red bar), which did not necessitate PLD2.

## Mechanical activation of TREK-1 by movement between nanodomains

As previously reported in a prior publication, we proposed that fluid shear induces the disassociation of PLD2 with cholesterol-dependent GM1 clusters and the association with $PIP_2$ clusters, leading to a change in spatial organization and activation of the enzyme (*Figure 1—figure supplement 1*; *Petersen et al., 2016*). Given that TREK-1 binds to PLD2, we hypothesized the channel could undergo a similar mechanically induced spatial reorganization between GM1 and $PIP_2$ clusters. To visualize this spatial patterning of TREK-1 in response to shear forces, we developed a method to chemically fix membranes during shear (*Figure 2A*). In our experiments, we employed cultured HEK293T cells expressing both TREK-1 and mPLD, which matched the conditions used in our electrophysiology experiments. These cells were subjected to shear forces of 3 dynes/cm$^2$, a physiologically relevant force to cells (*Petersen et al., 2016*; *Schneck, 1992*), using a rotary shear approach. Subsequently, the cells were fixed and labeled for two- or three-color dSTORM. TREK-1 proteins were tagged with EGFP, leveraging its inherent self-blinking properties for detection (*Call et al., 2023*). Lipids were labeled using Alexa 647 (A647) anti-$PIP_2$ antibodies or A647-labeled cholera toxin B (CTxB), which selectively bind to $PIP_2$ and GM1 clusters, respectively (*Petersen et al., 2016*). By concurrently labeling both lipids and proteins, we were able to monitor the dynamic changes in spatial organization within nanoscopic lipid domains (*Yuan and Hansen, 2023*). To access the $PIP_2$ domains on the inner leaflet, cells were permeabilized (see 'Materials and methods'). The effectiveness of cellular staining was confirmed through confocal microscopy (see *Figure 2—figure supplement 1A–G*).

Before the application of shear forces, TREK-1 and GM1 lipids displayed a robust correlation (*Figure 2B and C*), suggesting a close association between TREK-1 and GM1 lipids. However, after the application of shear forces, this correlation significantly diminished (p<0.01 at 50 nm). A similar experiment using a cy3b-labeled anti-TREK-1 antibody yielded nearly identical results, confirming the validity of both our EGFP-dSTORM method (*Call et al., 2023*) and the specificity of the TREK-1 antibody (*Figure 1—figure supplements 1A and 2A*).

Following the release of TREK-1 from GM1 lipids, we anticipated that it would undergo a nanoscale repositioning toward $PIP_2$ clusters. $PIP_2$ forms distinct nanodomains that are separate from GM1 clusters (*Petersen et al., 2016*; *van den Bogaart et al., 2011*; *Wang and Richards, 2012*) by an average distance of approximately 133 nm in HEK293T cells (*Yuan et al., 2022*). Cells were subjected to shear forces of 3 dynes/cm$^2$, permeabilized, fixed, and subsequently labeled with an anti-$PIP_2$ antibody. As predicted, the correlation between TREK-1 and $PIP_2$ was initially low prior to shear application, but it significantly increased after shear forces were applied (*Figure 2D*), in contrast to the observations made with GM1 (*Figure 2C*). Therefore, shear forces induce TREK-1 to disassociate from GM1 clusters and associate with $PIP_2$ clusters. Statistical analysis using a Student's *t*-test confirmed the significance of TREK-1 clustering with $PIP_2$ at both a single radius and multiple point comparisons along the curve.

Subsequently, we investigated the rearrangement of TREK-1 induced by shear forces in the presence of xPLD2. Like mPLD2, we overexpressed xPLD2 in HEK293T cells, mirroring the conditions employed in our electrophysiology experiments in *Figure 1B*. Furthermore, as with mPLD2, HEK293T cells were subjected to shear forces, permeabilized, labeled, and imaged with two-color dSTORM.

In the presence of xPLD2, shear caused TREK-1 to leave GM1 domains (*Figure 2E*). However, unlike the response observed with mPLD2, the association of TREK-1 with $PIP_2$ clusters remained relatively weak following shear forces (*Figure 2F*), despite an overall increase in TREK-1 and $PIP_2$ levels in the membrane prior to shear (*Figure 2—figure supplement 1C and D*). *Figure 2G* shows a model of shear-induced movement of TREK-1 from GM1 to $PIP_2$ clusters.

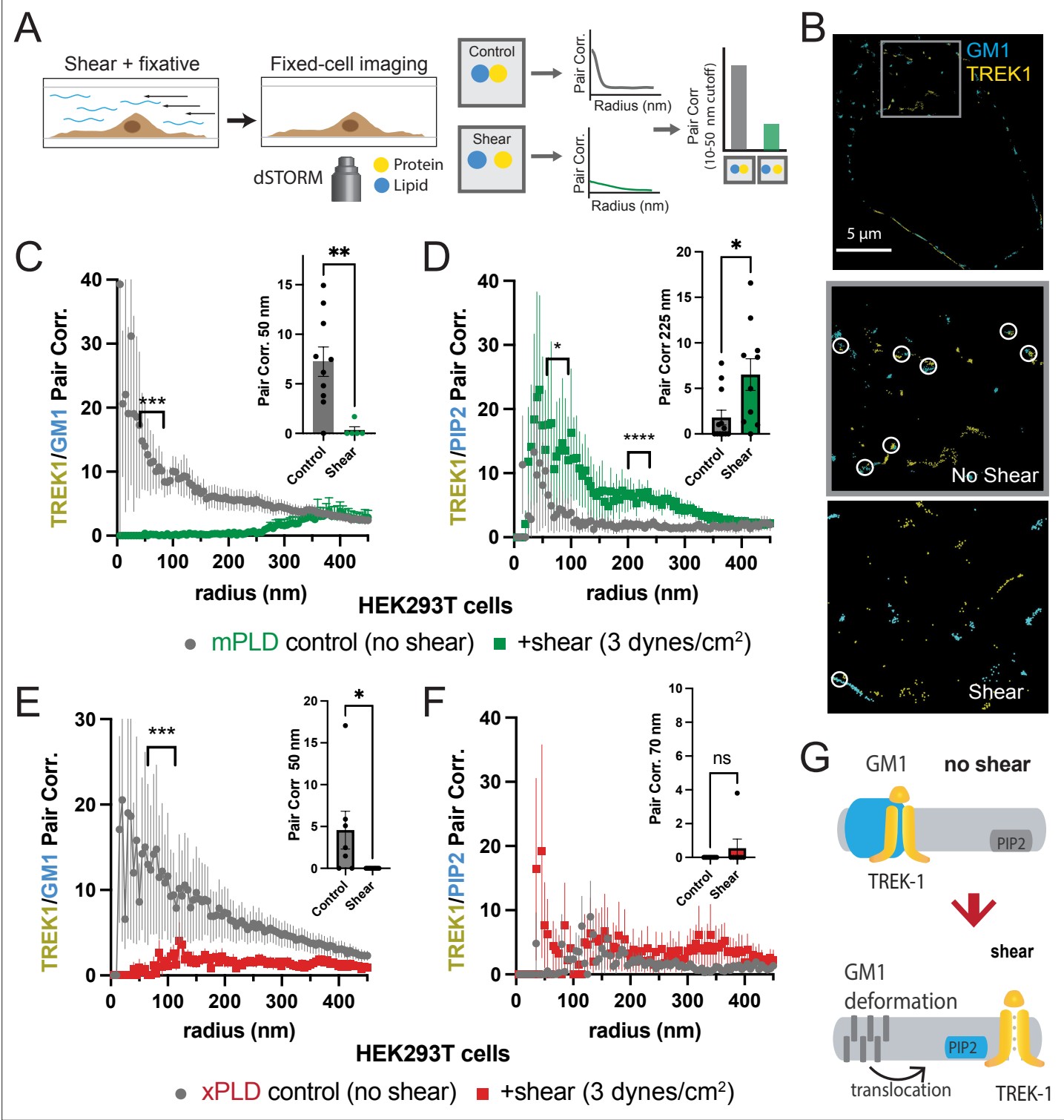

**Figure 2.** Shear induces nanoscopic movement of TREK-1 in HEK293T cells. (**A**) Schematic representation of the shear fixing protocol. Cells grown in a shear chamber are fixed while shear force is applied. Fixed samples are then labeled with fluorescent antibodies or CTxB and subjected to imaging for nanoscopic movement (<250 nm) by two-color super-resolution imaging and pair correlation (Pair corr.). (**B**) EGFP-STORM imaging of TREK-1:EGFP and Alexa 647 cholera toxin B (CTxB) with and without shear in HEK293T cells. The middle panel, outlined in gray, is a zoomed portion of the cell surface outlined in the top panel. The bottom panel is a zoomed portion of the cell surface from a cell treated with shear (see *Figure 2—figure supplement 1I* for full image). Locations of TREK-1/GM1 proximity are outlined with a white circle. (**C**) Pair correlation analysis (Pair corr.) of TREK-1 with GM1 lipids before and after shear (3 dynes/cm²; green) determined by EGFP-STORM imaging when mouse phospholipase D2 (mPLD2) is overexpressed

*Figure 2 continued on next page*

*Figure 2 continued*

(non-permeabilized). The significance of the Pair corr. change is shown across the range of radii 50–70 nm (along the curve) and at a single 50 nm radius (inset). (**D**) Combined EGFP-STORM imaging of TREK-1 with Alexa 647-labeled $PIP_2$ in the presence of overexpressed mPLD2 (permeabilized). Significance is shown for radii 70–85 nm along the curve and at a single 225 nm radius (inset). (**E, F**) Combined EGFP-STORM of TREK-1 in the presence of catalytically inactive PLD2 (xPLD2). Shear (3 dynes/cm²) of TREK-1 is shown as a red curve with xPLD2 present. The experiments are as described in panels (**C**) and (**D**). In (**E**) a significant shift in TREK-1/GM1 Pair corr. is shown for 50–70 nm (along the curve) and at a 50 nm radius (inset). In (**F**) Pair corr. did not appear to shift significantly, as determined by a Student's *t*-test or for multiple point a nested Student's *t*-test; *p<0.05, **p<0.01, ***p<0.001, ****p<0.0001. (**G**) Cartoon illustrating the association of TREK-1 with GM1 lipids prior to shear (top) and with $PIP_2$ lipids (bottom) in response to mechanical shear (red arrow).

The online version of this article includes the following figure supplement(s) for figure 2:

**Figure supplement 1.** Expression and staining of TREK-1 in cell culture.

**Figure supplement 2.** Comparisons of labeling type and permeabilization.

## Mechanism of PLD2 activation by shear

It is presumed that TREK-1 translocates between nanodomains as a complex with PLD2 (*Comoglio et al., 2014*). Lacking its own palmitoylation, TREK-1 interacts with PLD2 through its unstructured C-terminus. In earlier research, we demonstrated that cholesterol drives PLD2 associates with GM1 lipids. The application of mechanical force activated PLD2, presumably through the release of PLD2 from GM1 lipids. However, this PLD2 patterning under mechanical shear has not been directly demonstrated with dSTORM.

To validate our proposed mechanism regarding the shear-induced movement of PLD2 with dSTORM, we employed calibrated shear chambers (ibidi μ-Slide I⁰·⁴ parallel-plate) with cultured C2C12 muscle cells (mouse myocytes) and N2a mouse neuroblastoma cells, which naturally express TREK-1 (see *Figure 2—figure supplement 1E*). Utilizing endogenous expression helped circumvent potential artifacts that may arise from artificially saturating GM1 clusters through protein overexpression.

Our shear experiments were initiated by perfusing phosphate-buffered saline (PBS) through the calibrated shear chambers, applying a precisely controlled force of 3 dynes/cm². To maintain a constant temperature of 37°C, we employed a digitally controlled inline heater. Immediately following the application of shear forces (within <10 s), we introduced fixative agents into the shear buffer, facilitating the rapid fixation of cells in their mechanically stimulated state. This approach paralleled the methodology used in our experiments involving rotary shear (see *Figure 3A*, *Figure 3—figure supplement 1A*), but with tighter control of temperature.

Utilizing two-color dSTORM and analysis by the pair correlation function, we observed that shear forces introduce the mobilization of PLD2 within the cell membrane, and this effect was independent of temperature fluctuations. Prior to the application of shear forces, PLD2 exhibited a strong association with GM1 clusters (*Figure 3B*). However, after the application of shear forces, the correlation of PLD2 with GM1 clusters decreased significantly, while it robustly increased in association with $PIP_2$ clusters (*Figure 3C*). Notably, the release of PLD2 from GM1 domains induced by shear forces was more pronounced than the disruption caused by anesthetic agents, as previously reported (*Pavel et al., 2020*). The stability in temperature, maintained within a range of ±0.1°C, suggests the mechanism is unlikely to be attributed to the thermal melting of ordered lipids near a phase transition state.

Conducting cluster analysis of GM1 particles subjected to shear revealed a notable reduction in the apparent size of GM1 clusters, decreasing from 167 ± 3 to 131 ± 3 nm (approximately 20%, *Figure 3D*) in C2C12 cells. A similar effect on the size of GM1 clusters was also observed in N2a cells (*Figure 3E*). $PIP_2$ clusters also remained largely intact with a reduction in size (from 154 ± 1 to 139 ± 1) after shear (*Figure 3—figure supplement 1B*).

The observed reduction in cluster size closely resembled the outcomes of prior experiments in which we depleted cholesterol with methyl-beta-cyclodextrin (MBCD) (*Pavel et al., 2020*; *Petersen et al., 2016*). Given that MBCD is a known cholesterol removal agent, we conducted assays on N2a cells subjected to orbital fluid shear (3 dynes/cm²) to examine changes in cholesterol levels. The cells were subjected to 10 min of shear at 37°C, followed by fixation.

Our results revealed a 25% reduction in free cholesterol levels in N2a cells exposed to orbital shear (p<0.001) (*Figure 3F*), which coincided with the activation of PLD (*Figure 3G*). Importantly, this decrease in cholesterol was statistically significant p<0.0001 and reversible. Allowing the cells to

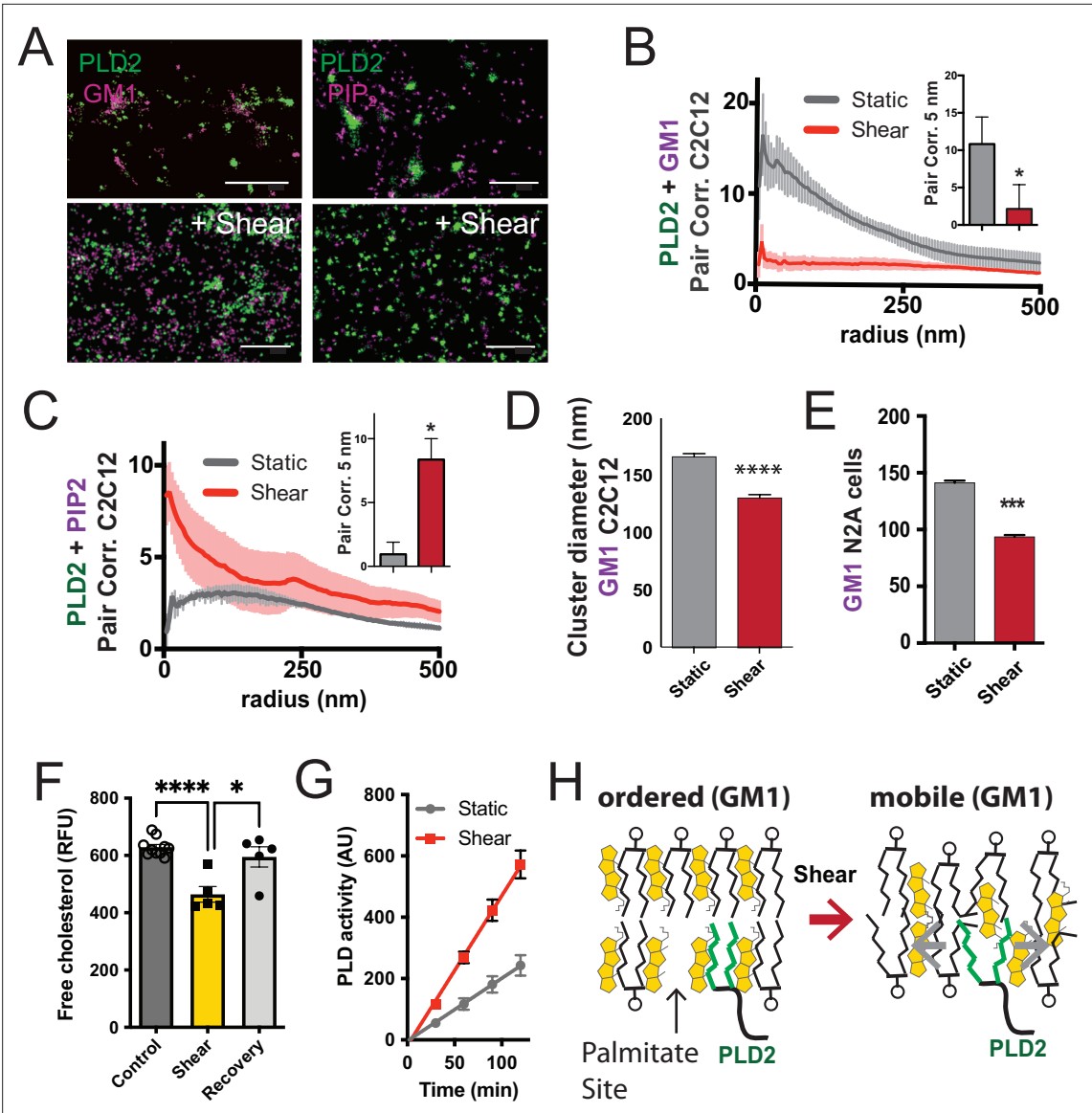

**Figure 3.** Shear mobilizes PLD2 within ordered GM1 lipids. (**A**) Two-color dSTORM images of fixed C2C12 cells with and without (3 dynes/cm²) shear. Cells were labeled with fluorescent CTxB (ganglioside GM1) or antibodies (anti-PIP₂ or anti-PLD2 as indicated) and sheared with the temperature held constant at 37°C. Scale bar = 1 μm. (**B**) Pair correlation analysis (Pair corr., unitless) of PLD2 with GM1 or PIP₂ lipids at a given radius. Error bars are displayed at a given radius. A bar graph (inset) is shown at the shortest calculated radius of 5 nm (single point on the x-axis). Prior to shear (gray line), PLD2 associates with GM1 clusters; after shear, there is almost no association. (**C**) The opposite was true for phosphatidylinositol 4,5 bisphosphate (PIP₂). Prior to shear, PLD2 does not associated significantly with PIP₂ clusters, after shear, its association increases dramatically. (**D**) Cluster analysis of the GM1 lipids from the C2C12 cells shown in (**A**). (**E**) Cluster analysis of GM1 lipids in neuroblastoma 2a (N2a) after 3 dynes/cm² shear force. (**F**) Fluorescent cholesterol assay. N2a cells grown in 48-well plates were sheared with 3 dynes/cm² orbital fluid shear, fixed with shear (10 min), and compared to control cells with no shear using a fluorescent cholesterol assay. After shear, a second set of control cells were allowed to recover with no shear and fixative for 30 s (recovery), otherwise the cells were treated identical to experimental cells (n = 5–10). (**G**) A live PLD activity assay demonstrates that fluid shear (3 dynes/cm²) increases substrate hydrolysis in cultured N2a cells (n > 800 clusters from 5 to 6 cells). (**H**) Depiction of shear thinning activating PLD2. Left: the palmitates of PLD2 (green lines) are shown bound to the palmitate site in ordered GM1 lipids. Right: after shear cholesterol is reduced, and GM1 lipids are deformed. The deformed surface no longer binds palmitates efficiently allowing the palmitates move freely—a process known as shear thinning. Statistical comparisons were made with an unpaired Student's *t*-test (*p<0.05, ***p<0.001, ****p<0.0001).

The online version of this article includes the following figure supplement(s) for figure 3:

**Figure supplement 1.** Observing cellular changes in response to mechanical stimulation.

recover for approximately 30 s prior to fixation resulted in the restoration of cholesterol levels to those observed in non-sheared cells (*Figure 3F*).

In C2C12 cells, TREK-1 and PLD2 exhibited a strong correlation both before (*Figure 3—figure supplement 1C*, gray trace) and after shear (*Figure 3—figure supplement 1C*, red trace), providing further evidence that they form a complex at least intermittently, in both shear and unsheared states. Following shear, a minor fraction of TREK-1 was found to associate with $PIP_2$ (*Figure 3—figure supplement 1D*). Interestingly, in contrast to HEK293T cells, C2C12 cells displayed minimal TREK-1/ GM1 correlation (*Figure 3—figure supplement 1E*, gray trace). Notably, PLD2 regulation is known to be influenced by cholesterol (*Petersen et al., 2016*). In various cell types, we have observed that cultured cells tend to have lower cholesterol levels than human tissues (*Wang et al., 2023*; *Wang et al., 2021*; *Yuan et al., 2022*). These observations lead us to consider the possibility of introducing more physiological levels of cholesterol in our experimental setup for cultured cells.

## Regulation of TREK-1 clustering by cholesterol and GM1 lipids

Cholesterol levels, especially in the brain, can be notably high (*Hansen, 2023*; *Zhang and Liu, 2015*). We hypothesized that cholesterol might influence the endogenous TREK-1 to associate with PLD2 in GM1 clusters. To test this hypothesis, we introduced cholesterol into C2C12 cells using apolipoprotein E (apoE) lipidated with 10% serum as apoE is a naturally occurring cholesterol transport protein (*Wang et al., 2023*; *Wang et al., 2021*; *Yuan et al., 2022*; see *Figure 4A*). Remarkably, lipidated apoE induced a substantial clustering of TREK-1 with GM1 lipids in membranes of C2C12 cells (*Figure 4B*). This effect was also observed with TREK-1 in N2a cells. Importantly, the application of fluid shear (3 dynes/cm$^2$) completely reversed the effect of cholesterol (*Figure 4C*).

We anticipated that the cholesterol-induced association of TREK-1 with GM1 lipids would lead to a reduction in TREK-1 currents as PLD2 would be inhibited due to a lack of substrate. To directly assess the activity of TREK-1 under conditions of elevated cellular cholesterol, we overexpressed TREK-1 in HEK293T cells and quantified the current density in whole-cell patch-clamp mode, both with and without cholesterol uptake (*Figure 4D*). Consistent with our proposed model, TREK-1 current density decreased nearly 2.5-fold in cholesterol-loaded cells, and this reduction in current was statistically significant (p<0.05).

In our earlier investigation, we established that astrocytes play a pivotal role in regulating the clustering of proteins in neurons. This regulation is achieved through the release of apoE-containing particles, which are primarily laden with cholesterol. Remarkably, our studies revealed that cholesterol amplifies the clustering of proteins within GM1 domains. Furthermore, we demonstrated that disrupting the cholesterol synthesis in astrocytes leads to the reversal of protein clustering, as previously reported (*Wang et al., 2021*). It stands to reason that the same astrocytic cholesterol dynamics could govern the localization of TREK-1 within GM1 lipids in the brain of a mouse.

To investigate the in vivo regulation of TREK-1 clustering by astrocytes, we stained 50 micron brain slices obtained from both control mice and mice specifically engineered to lack cholesterol synthesis in astrocytes. The depletion of cholesterol from astrocytes was achieved by targeting sterol regulatory element-binding protein 2 (SREBP2), a crucial regulator of cholesterol synthesis (*Ferris et al., 2017*; *Wang et al., 2021*). Before slicing, the brains underwent fixation via whole-body perfusion. The resulting free-floating brain slices were mounted onto circular cover slips using a fiberglass filter paper, which facilitated dSTORM buffer access to the tissue with minimal background interference (see *Figure 2—figure supplement 2E*). These brain slices, taken from coronal sections near the hippocampus, were subjected to staining with cy3b-anti-TREK-1 antibody and A647-CTxB. The imaging process encompassed multiple unspecified locations (approximately 10 regions), and subsequent analysis was performed employing the pair correlation function.

In flox control mice (wild-type SREBP2), TREK-1/GM1 correlation exhibited a distinct pattern, with TREK-1 clustering near GM1 (*Figure 4E*), aligning with the GM1 correlation seen in N2a cells with elevated levels of cholesterol (*Figure 4B*). As anticipated, in the brains of mice with astrocyte-specific SREBP2 deficiency, TREK-1 displayed a reduced association with GM1 lipids (p<0.0001). While highly variable, these findings serve as evidence that, in the brain of an animal, astrocytic cholesterol serves as a critical regulator of TREK-1's affinity for inhibitory GM1 clusters.

*Figure 4F* presents a comprehensive model detailing the regulation of TREK-1 by astrocyte-derived cholesterol, using the insights derived from both cultured cells and ex vivo studies. In this

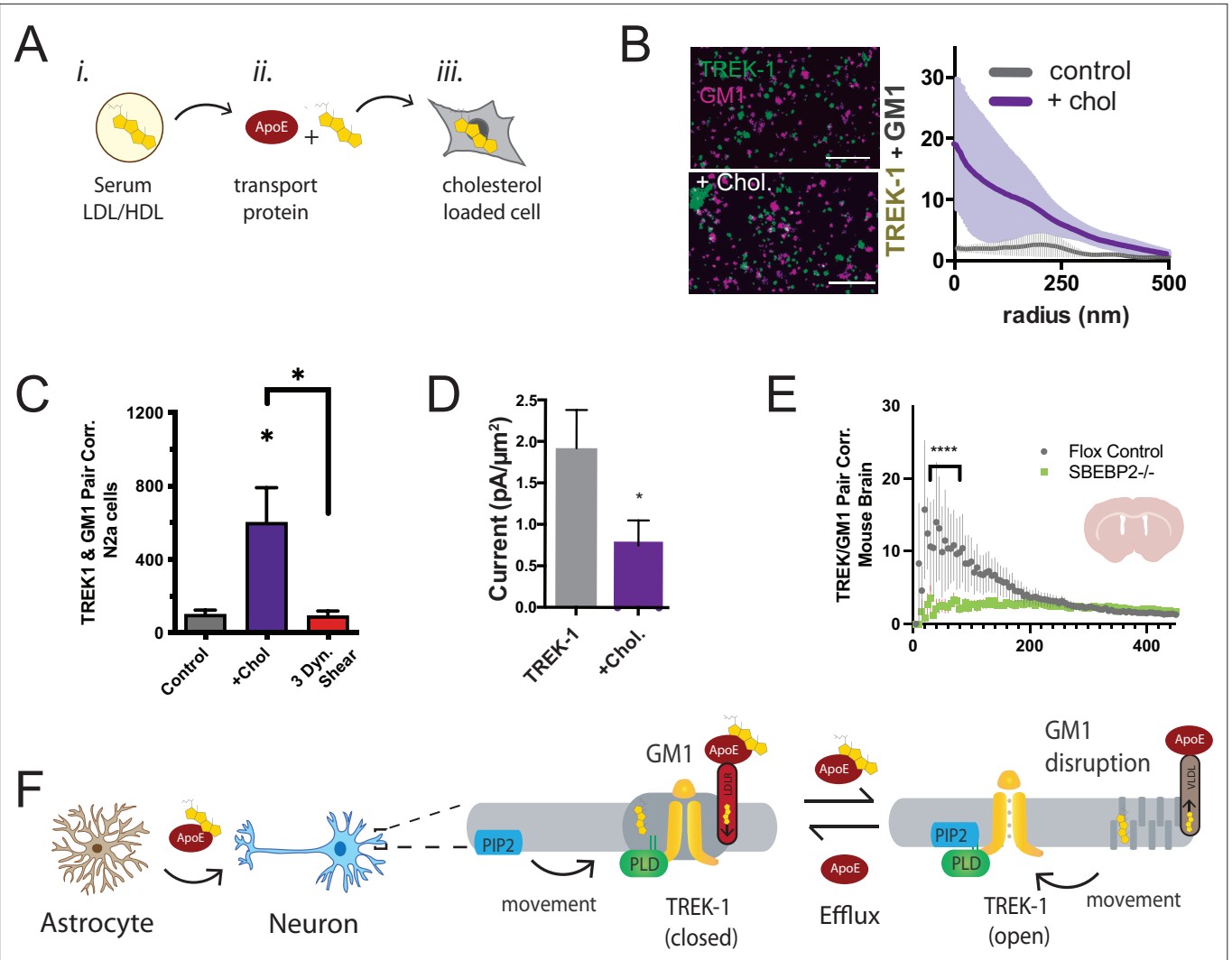

**Figure 4.** Astrocyte cholesterol regulates TREK-1 through GM1 lipids and spatial patterning. (**A**) Uptake of cholesterol into cultured cells using the cholesterol transport protein apolipoprotein E (apoE). (**B**) Cholesterol/lipid uptake into C2C12 cells with 4 µg/ml (~110 nM apoE, purple line). Cholesterol dramatically increases TREK-1 correlation of TREK-1 with GM1-labeled lipids. Without cholesterol (gray line) very little TREK-1 clusters with GM1 lipids. Scale bars = 1 µm. (**C**) Pair correlation (Pair corr.) of TREK-1 and CTxB, localized within 5 nm of each other, are shown plotted after cholesterol treatment (apoE + serum, purple shading) or treatment with 3 dynes/cm² shear (red shading). Cholesterol increased TREK-1 association almost fivefold and shear reversed the effect (n = 4–7); unpaired Student's *t*-test. (**D**) Current densities from whole-cell patch-clamp recordings are shown with and without cholesterol loading with 4 µg/ml apoE in HEK293T cells over expressing human TREK-1. Increasing cholesterol inhibited the channel approximately threefold (Student's *t*-test; *p<0.05) (**E**) Reduction in neuronal cholesterol results in a decrease in correlation between TREK-1 and GM1 cluster in brain slices from a hGFAP-Cre driving the SB2 knockout mouse and its Flox control. Student's *t*-test at 25 nm *p,0.05, nested Student's *t*-test at 25–50 nm (****p<0.0001, n = 20–24 unspecified cortical regions). (**F**) Proposed model for cluster associated TREK-1 activation and inhibition. In high cholesterol, TREK-1 clusters with PLD2 and ordered (thick) GM1 lipids inhibiting the channel. In low cholesterol, TREK-1 is partially clustered closer to PIP₂ generating basal TREK-1 activity (see also *Figure 1C*). After shear, the order of GM1 clusters (dark gray) is disrupted further increasing PLD2 and TREK-1 clustering with PIP₂ lipids (blue).

The online version of this article includes the following figure supplement(s) for figure 4:

**Figure supplement 1.** Proposed model for domain-mediated mechanosensation of TREK-1 by spatial patterning.

model, cholesterol originating from astrocytes is transported to neurons via apoE. Under conditions of elevated membrane cholesterol, TREK-1 forms associations with inhibitory GM1 lipids, a scenario in which PLD2 has limited substrate availability. Conversely, when membrane cholesterol levels are reduced, TREK-1 relocates away from GM1 lipids toward activating PIP₂ lipids. In this context, PLD2 gains improved access to its substrate phosphatidylcholine (PC), leading to the production of lipid

agonists such as PA, which ultimately evoke activation of the TREK-1 channel (see also *Figure 4—figure supplement 1*).

## Direct activation of PLD2 by osmotic stretch

Osmotic stretch is a well-documented activator of TREK-1 (*Patel et al., 1998*). To investigate whether osmotic stretch also activates PLD2, we conducted experiments to monitor PLD product release in live cells subjected to hypotonic stretch. Our findings revealed that membrane stretch by a 70 mOsm swell (representing a high degree of stress) led to approximately 50% increase in PLD2 activity in N2a cells (*Figure 5A*). It is worth noting that this level of activation, although significant, was less pronounced compared to shear stress in N2a cells (*Figure 3E*), which increased approximately threefold in activity.

Subsequently, we conducted dSTORM experiments to investigate whether stretch had the capacity to release PLD2 from GM1 lipids within the membranes of N2a cells. In these experiments, cells were exposed to either a hypotonic solution with an osmolality of 70 mOsm (indicating a state of swelling) or an isotonic solution with an osmolality of 310 mOsm (representing a control condition) for 15 min at 37°C. Following this treatment, the cells were fixed, labeled, and subjected to imaging using the same procedures employed in the shear stress experiments. Our results, consistent with the outcomes of shear stress experiments and PLD2 assay, clearly demonstrated a discernible shift in the spatial distribution of PLD2, transitioning from ordered GM1 clusters to the $PIP_2$ clusters after osmotic stretch (*Figure 5B–D*).

## PA regulation of mechanosensitivity thresholds in vivo

In order to gain deeper insights into the in vivo role of PLD2 in mechanotransduction, we conducted investigations into mechano-thresholds and pain perception in *Drosophila*, employing a PLD-knockout model (*Thakur et al., 2016*). It is worth noting that, while the specific downstream effectors of PLD in fruit flies remain unidentified, and they possess only distantly related TREK-1 orthologs, (as discussed in a later section), the utilization of fruit flies as a model organism is advantageous because they possess a single PLD gene (*dPLD*).

Our analysis revealed the presence of GM1 domains in dissected fly brains, and we observed that dPLD exhibited a robust response to shear stress of 3 dynes/cm$^2$ when cultured in neuronally derived fly cell line, BG2-c2 (see *Figure 6A and B*). The activity of PLD increased by nearly fourfold in the fly cell line (*Figure 6B*), a result consistent with the response of PLD to mechanical force observed in HEK293T cells (*Figure 3G*).

Subsequently, we preceded to examine the in vivo role for PA in mechanosensation employing single-animal measurements of arousal threshold (*Murphy et al., 2017*; *Murphy et al., 2016*; see *Figure 6—figure supplement 1A*). The arousal assay quantifies the amount of mechanical stimulation required to elicit movement from a resting fly. Over a period of 24 hr, the flies were exposed to a series of incremental vibrational stimuli every 30 min. For each series, the threshold of stimulation necessary to induce movement in the fly, as indicated by observable motion, was recorded using automated machine vision tracking. These recorded measurements were then compared to genetically matched control groups.

PLD[null] flies exhibited a notably lower arousal threshold compared to their control counterparts (*Figure 6C*). This lower arousal implies an increased sensitivity to mechanical force. Furthermore, we employed a neuronal-specific driver, Nsyb GAL4, in conjunction with a PLD RNAi line (PLD-KD) to investigate the role of PLD in the central nervous system. The neuronal knockdown of PLD yielded a similar increase in mechanosensitivity, providing clear evidence that this phenotype is specific to neuronal functions (*Figure 6D*).

Furthermore, we conducted assessments to determine the role of PLD in fly shock avoidance, serving as a measure of an adverse electrical stimulus (*Figure 6—figure supplement 1B*). To assess their responses, PLD-KD flies were positioned at the choicepoint of a T maze, where they were given the option to select between a chamber inducing a noxious shock or a non-shock control chamber (*Drago and Davis, 2016*). Flies were subjected to incrementally increasing voltages of electrical shock. Our findings revealed that PLD[null] flies exhibited heightened sensitivity to electric shock compared to control groups (*Figure 6E and F*). In *Figure 6G*, we present a potential model illustrating how PA may contribute to mechanosensitivity in a fly.

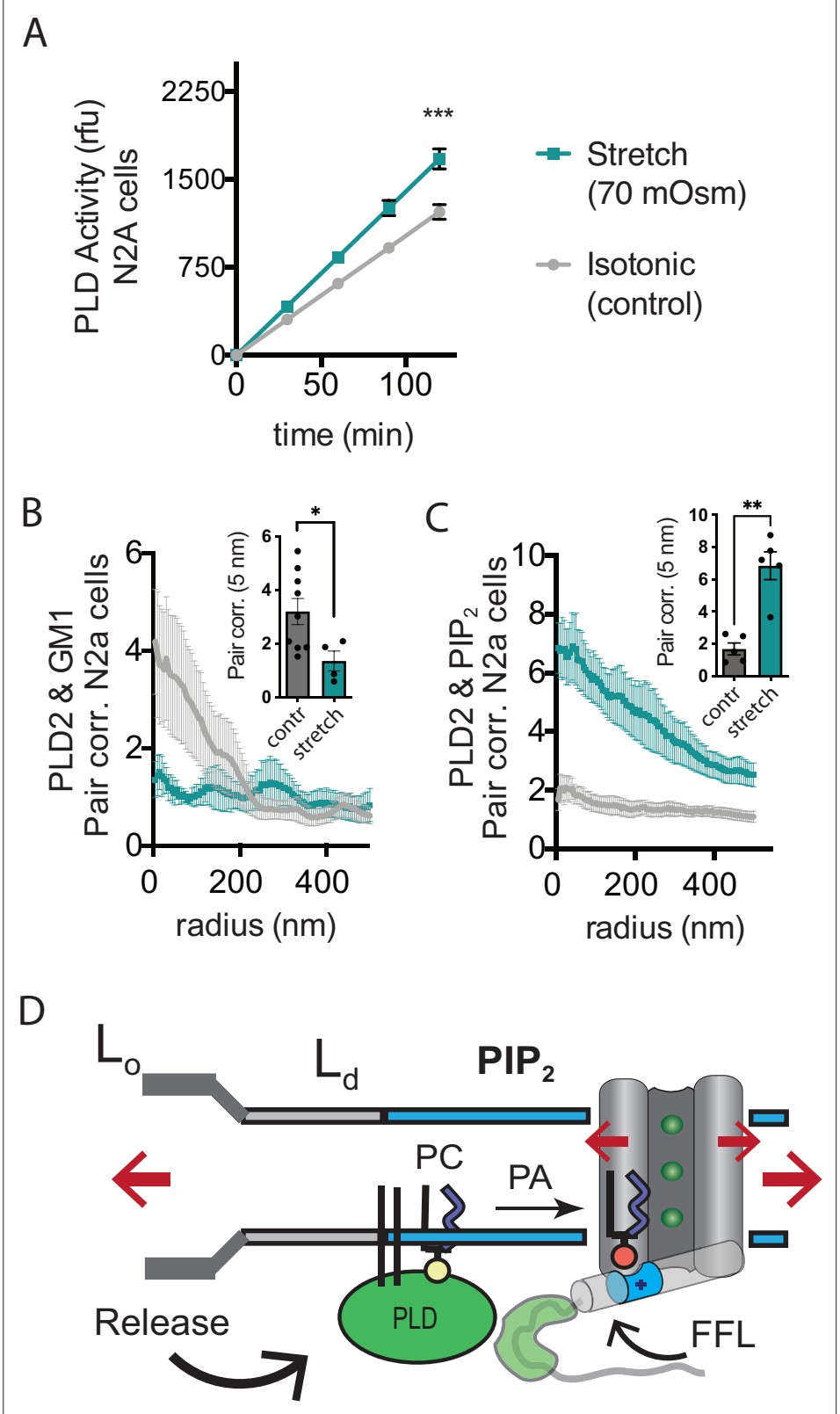

**Figure 5.** Osmotic stretch activates PLD2 in N2a cells. (**A**) Stretch by osmotic swell (70 mOsm buffer) increased PLD2 catalytic activity in live neuroblastoma 2a (N2a) cells compared to isotonic control cells (310 mOsm) (n = 5). (**B, C**) dSTORM imaging showing PLD2 trafficking from ganglioside (GM1) to phosphatidylinositol 4,5-bisphosphate ($PIP_2$) clusters in response to 70 mOsm stretch in N2a cells (coloring the same as in panel

*Figure 5 continued on next page*

*Figure 5 continued*

**A**). Cells were treated, fixed, and labeled with anti-PIP$_2$ or PLD2 antibody or cholera toxin B (CTxB, GM1). Prior to stretch, PLD2 clustered with GM1 lipids and very little with PIP$_2$ (gray lines). After stretch PLD2 clustered robustly with PIP$_2$ but very little with GM1 lipids. Insets show the change in correlation at 5 nm, Student's *t*-test (n = 5–9), *p<0.05, **p<0.01, ***p<0.001. (**D**) Summary figure for the proposed combined model of TREK-1 in a biological membrane by evoked PLD2-dependent currents and mechanical activation by direct force from lipid (FFL). TREK-1 in the open conformation (green ions), in complex with PLD2, is shown after stretch (large red arrows) associating (curved black arrow) with PIP$_2$ clusters (blue shaded bars) in the thin liquid disordered (Ld) region of the membrane (light gray). A known gating helix (gray cylinder) is shown in the up (open channel) position with a PLD2 binding site immediately following the helix (green tube). The opening is a response to three factors that combine to raise the gating helix to the up position. (1) FFL (small red arrows) in TREK-1 favors an open (up) helix conformation. (2) The tip is brought into proximity of PLD2 in the open position and (3) PA (red sphere) is produced and maintains the up positioned by binding to charged residues (blue tube) pulling the helix toward the membrane.

## Discussion

In summary, our findings collectively demonstrate that the spatial distribution of TREK-1 and PLD2 in association with GM1 and PIP$_2$ lipids, along with the generation of PLD2-derived PA, is essential for eliciting the complete mechanically evoked current from TREK-1 in a biological membrane. Furthermore, the disruption of PLD2 localization within a lipid nanodomain provides a compelling explanation for how the C-terminus confers mechanosensitivity to a channel, particularly when the domain lacks structural integrity.

In contrast, the activation of purified TREK-1 in soy PC does not necessitate the involvement of the C-terminus, as indicated by prior research (*Brohawn et al., 2014b*). This observation implies the existence of a PLD2-independent mechanism of action. The precise relative contributions of the two mechanisms in endogenous tissue remain unclear. Notably, in HEK293T cells, a current density of 0.05 pA/cm$^2$ current density (representing less than 10% of the total stretch current) was found to be PLD2 independent (see *Figure 1C* inset). This observation raises the possibility that the localization of TREK-1 to mechanosensitive lipids essential for its activation might be lacking in this context, or alternatively, the propagation of tension within the plasma membrane might differ. Consequently, there is a need for a more comprehensive understanding of TREK-1 spatial patterning within specific cell types, coupled with an understanding of stretch-induced tension in the different lipid nanodomains. Furthermore, it is plausible that a portion of the PLD2-dependent stretch current merely augments the direct mechanosensitivity of TREK-1. However, it is important to note that this enhancement mechanism is not obligatory as the channel is directly activated by PA in purified lipids absent stretch and tension (*Cabanos et al., 2017*). This can also explain the discrepancies between liposome-based mechanosensitivity since crude soy extracts (azolectin) are known to contain anionic lipids, thus may negate the need for a C-terminus and providing the PA needed for full TREK activation.

As mentioned previously, PLD2 is a soluble enzyme that associates with the membrane through palmitate. Given its lack of a transmembrane domain, mechanisms of mechanosensation involving hydrophobic mismatch, tension, midplane bending, and curvature can be largely ruled out, leaving the kinetic mechanism as the most plausible. The palmate moiety establishes van der Waals bonds with the GM1 lipids. The disruption of these bonds is necessary for PLD to disassociate from GM1 lipids, suggesting a mechanism akin to 'shear thinning' within GM1 clusters. Shear thinning is a term in rheology that describes the phenomenon wherein viscous mixtures become less viscous in response to shear or stretching forces (*Küçüksönmez and Servantie, 2020*). This process is inherently kinetic in nature and operates by mechanically disrupting noncovalent bonds, thereby allowing the molecules to move relative to one another (see *Figure 3H* and *Figure 3—figure supplement 1F* for detailed description).

The spatial patterning of TREK-1 with PIP$_2$ clusters may depend on the channel's conformational state, as evidenced by the observation that xPLD/TREK-1 combination exhibited reduced associated with PIP$_2$ after shear compared to the mPLD/TREK-1 combination (*Figure 2D and F*) despite an increase in the levels of PIP$_2$ and TREK-1 expression prior to shear (*Figure 2—figure supplement 1C–D and G*). This observation aligns with the up-down model of the gating helix (*Brohawn et al., 2014a*). As illustrated in *Figure 5D*, we propose a theoretical mechanism that accounts for the coordinated signaling involving PA and for combined PA signaling and direct force from lipid

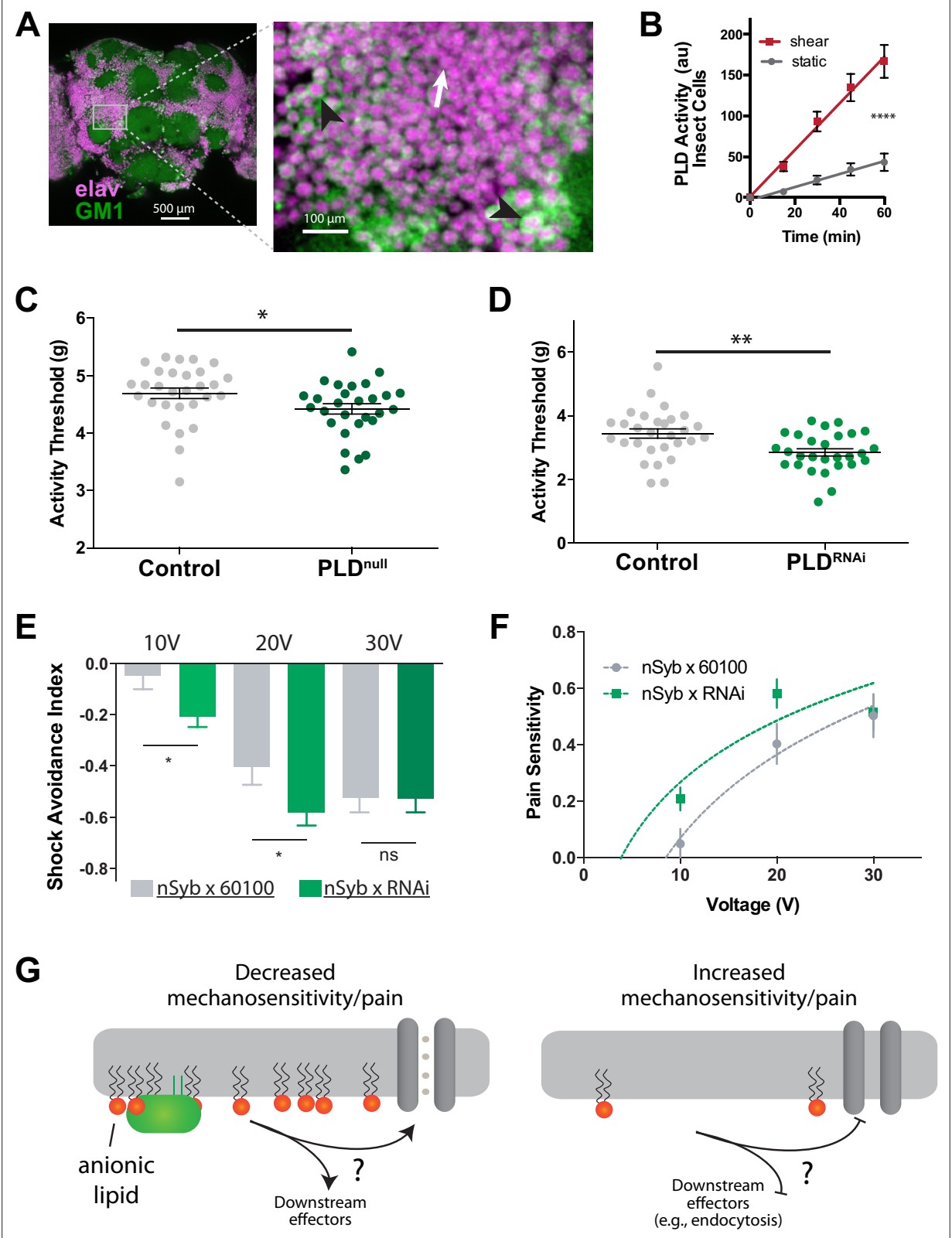

**Figure 6.** PLD modulates mechanosensitivity in *Drosophila*. (**A**) Cholera toxin B (CTxB) robustly labels GM1 lipids (GM1, green) throughout the brain of *Drosophila* (left). The zoomed section (right) shows that most of the labeling is found on the membrane. There are notable variations in the amount of CTxB labeling, with some cells expressing GM1 over the entire membrane (black arrows) while others only have labeling in small puncta (white arrow). (**B**) Shear (3 dynes/cm²) robustly activates PLD2 in a live PLD assay with cultures neuronal insect cells. (**C**) Measurements of *Drosophila* mechanosensation

*Figure 6 continued on next page*

*Figure 6 continued*

in vivo. Animals with or without the pld$^{null}$ gene were stimulated by increasing amounts of mechanical vibration (see *Figure 6—figure supplement 1*). Flies lacking PLD2 had a decreased threshold (i.e., more sensitivity to mechanical stimulation) compared to genetically matched controls (w1118) (p=0.02, n = 28–29), consistent with the prediction that PA decreases excitability of nerves. (**D**) The same result was observed in a PLD$^{RNAi}$ line which results in PLD knockdown only in the neurons of *Drosophila* (p=0.002, n = 28–29), Mann–Whitney test. (**E, F**) Flies were subjected to increasing voltages of electrical shock in a two-choice assay. PLD-KD flies showed an increased sensitivity to shock when compared with wild-type flies. PLD$^{RNAi}$ flies had a higher aversion to shock at 10 V (p=0.0213, n = 21) and 20 V (p=0.0492, n = 27–30), but not at 30 V (p=0.672, n = 12). (**G**) Proposed role of PLD2 in regulating mechanical thresholds. PA is a signaling lipid in the membrane that activates hyperpolarizing channels and transporters. When PA is low the membrane is less polarized, and cells are more sensitive to mechanical activation. The downstream targets are unknown (shown with a '?'). Flies lack a known mechanosensitive TREK-1 homolog.

The online version of this article includes the following figure supplement(s) for figure 6:

**Figure supplement 1.** Pan-neuronal knockdown of *pld* in *Drosophila* alters sensitivity to electrical shock.

in a biological membrane. This mechanism suggests that conformational changes conducive to the helix-up (open) position are sustained only when PLD2 is present and capable of producing localized PA. In the absence of PA, PIP$_2$ forces the channel closed (*Cabanos et al., 2017*; *Chemin et al., 2007b*) and the helix in the down position (*Figure 1—figure supplement 2E and F*).

In control experiments that estimated protein levels through fluorescent labeling, we observed a significant decrease in TREK-1 protein levels after fixation with mPLD2 or shear for 15 min (*Figure 2—figure supplement 2G*). While it is conceivable that this decline in protein levels might have occurred in less than 2 ms, that is, faster than we are able to observe (see *Figure 1—figure supplement 2D*), which could explain the loss of TREK-1 current in *Figure 1B*, independent of local PA gating of TREK-1, endocytosis, the process that removes TREK-1 from the plasma membrane is not likely to occur in less than 2 ms (*Soykan et al., 2017*). Furthermore, the reduction in TREK-1 levels was comparable when wt. PLD2 was employed (*Figure 2—figure supplement 1G*), suggesting that alterations in protein levels are unlikely to account for the diminished TREK-1 pressure currents observed with xPLD2 in *Figure 1B*.

It is important to note that PLD2 is recognized for its ability to enhance endocytosis (*Du et al., 2004*). Specifically, xPLD2 has been shown to inhibit both agonist-induced and constitutive endocytosis of μ-opioid receptor, effectively retaining the receptor on the membrane (*Koch et al., 2006*). The phenomenon of membrane retention is not unique to the μ-opioid receptor but extends to many proteins, including Rho (*Wheeler et al., 2015*), ARF (*Rankovic et al., 2009*), and ACE2 (*Du et al., 2004*), among others. Consequently, it is not surprising that the overexpression of xPLD resulted in an increase in TREK-1 surface expression (*Figure 2—figure supplement 1C and G*). Conversely, the overexpression of mPLD led to the lowest levels of TREK_1 expression, which aligns with its propensity for promoting endocytosis (*Figure 2—figure supplement 1G and H*). Given that mPLD2 yielded the highest mechanically evoked TREK-1 current despite the lowest protein levels, it appears unlikely that protein expression levels are the limiting factor governing mechanically evoked TREK-1 currents in HEK293T cells.

The latency of PLD2-dependent activation is important as it offers insights into the potential physiological processes in which PLD2 may play a role in mechanotransduction. Drawing from estimates involving diffusion from GM1 to PIP$_2$, we estimated a latency of ~650 μs (*Petersen et al., 2016*). To establish an upper temporal limit for PLD2 to become activated and generate PA in proximity to TREK-1, we leveraged the rate of increase in TREK-1 activity observed with WT PLD2. We observed initial TREK-1 currents almost immediately and these currents became notably significant within 2.1 ms at a pressure of 60 mmHg (*Figure 1—figure supplement 2D*). This time frame is highly likely to be faster than the margin of error associated with our instrument setup. While we did not perform a precise calibration of our setup's error, conservative estimates based on manufacturer specifications suggest an upper limit of approximately 10 ms.

The requirement for both PLD2 activity and the C-terminus to activate TREK-1 by pressure provides further validation for the previously proposed conclusion that TREK-1 is gated by a local high concentration of PA (*Cabanos et al., 2017*; *Comoglio et al., 2014*). In theory, PLD2 activity could elevate global PA levels to a point where TREK-1 becomes activated without the need for spatial patterning through protein–protein interactions. However, our experiments with truncated TREK-1 and 60 mmHg pressure in HEK293T cells did not support this hypothesis (*Figure 1C*).

Our findings regarding the mechanism we identified remained consistent across cell types (HEK, N2a, C2C12, mouse neurons) despite their significant differences in cellular characteristics. We were able to induce protein–lipid domain associations in all these cell types through the manipulation of cholesterol levels. This intriguing observation suggests a potential avenue of research to investigate whether cholesterol or other lipids undergo significant changes at sites where mechanosensation is known to occur, such as at the termini of low- or high-threshold mechanoreceptors (*Handler and Ginty, 2021*). It is conceivable that various neuronal types and the channels localized within them adapt to different types of mechanosensation by modulating the lipid environment where sensory perception is most critical. For instance, inflammation-induced cholesterol uptake (*Hansen and Wang, 2023*) might sensitize channels to pain perception (*Petersen et al., 2020*).

Additionally, our experiments with flies highlight the importance of the upstream effectors in these systems. While the specific binding partners for PLD2 and its potential downstream effects in flies remain unknown, the observed phenotypes strongly suggest that PLD plays a crucial role in normal sensory perception, which appears to be more evolutionarily conserved than TREK-1. It is possible that a single lipid or a mechanism regulating multiple channels simultaneously may have a more significant impact compared to the modulation of a single channel.

The regulation of mechanosensation and electric shock responses in *Drosophila* by PA (*Figure 6*) provides in vivo support for the growing understanding of the role of anionic lipids in setting force-sensing thresholds. For example, $PIP_2$ sets the threshold for mechanical B-cell activation (*Wan et al., 2018*), while sphingosine-1-phosphate (S1P), a lipid similar to PA, regulates pain in mice (*Hill et al., 2018*). The activation of PLD2 by mechanical force and substrate presentation helps elucidate how an enzyme could directly set pain thresholds and mechanosensitivity via canonical mechanosensitive ion channels and endocytosis.

Palmitoylation, a modification found in many important signaling molecules including tyrosine kinases, GTPases, CD4/8, and almost all G-protein alpha subunits (*Aicart-Ramos et al., 2011*), can target proteins to GM1 domains (*Leventat et al., 2010*). The spatial patterning of these palmitoylated proteins or their binding partner by mechanical force may alter the availability of substrates and affect downstream signaling. Spatial patterning, along with its effect on palmitoylated G-proteins, likely contributes to the mechanosensitivity observed in many GPCRs (*Gudi et al., 1998*; *Storch et al., 2012*; *Wei et al., 2018*; *Xu et al., 2018*). Many ion channels are also palmitoylated (*Shipston, 2011*). For example, the voltage-gated sodium channel ($Na_v$)1.9 clusters with GM1 lipids, and disrupting this clustering (e.g., by chemically removing cholesterol) induces a corresponding pain response (*Amsalem et al., 2018*; *Wan et al., 2018*). Similarly, S1P, an anionic lipid similar to PA, regulates pain in mice (*Hill et al., 2018*).

It is worth noting that $PIP_2$ gates many ion channels (*Cabanos et al., 2017*; *Chung et al., 2019*; *Hansen, 2015*; *Hansen et al., 2011*; *Robinson et al., 2019*), and the spatial patterning induced by $PIP_2$ should be considered in combination with the direct effect of the lipid on the channel. Cholesterol, known to bind directly to and modulate channels (*Levitan et al., 2014*), should also be considered separately from its influence on spatial patterning through GM1 lipids. These direct lipid effects are likely to interact with spatial patterning to collectively regulate the biochemical processes within a cell.

Lastly, in our study, we utilized cholera toxin B (CTxB) and antibody probes for lipid labeling, recognizing that these lipid probes have previously been shown to induce artificial clustering in unfixed cells, a phenomenon referred to as 'antibody patching' (*Moon et al., 2017*; *Petersen et al., 2020*; *Veatch et al., 2012*). Antibody patching has historically served as a useful tool to segregate proteins within biological membrane, facilitating the determination of their precise localization (*Lang et al., 2001*; *Yuan and Hansen, 2023*). In the context of our research, we anticipate that antibody patching might similarly affect lipid distribution, offering a valuable advantage in clearly elucidating the co-localization of proteins with lipids in the membrane.

To address concerns related to potential artifacts stemming from clustering, we adopted a two-step fixation procedure. Initially, cells were fixed during treatment to minimize lipid mobility, followed by a second fixation step after labeling, ensuring fixed antibodies and high localization precision during dSTORM. Additionally, we assessed lipid diffusion after fixing using our typical experimental protocols. In our system, combined paraformaldehyde and glutaraldehyde effectively inhibited GM1 lipid diffusion, as confirmed by fluorescence recovery after photobleaching (FRAP) experiments conducted in live cells (*Figure 3—figure supplement 1G and H*).

It is also worth considering potential artifacts arising from variations in labeling density and overcounting, which could impact the size of the lipid clusters analyzed in *Figure 3*. We quantified the amount of fluorescent labeling (*Figure 3—figure supplement 1H*) and observed a slight reduction in GM1 and PIP$_2$ labeling after shear, although this change did not reach statistical significance.

The mechanical disruption of PLD2 with GM1 lipids and its association with PIP$_2$ was determined by the pair correlation function, a method that is robust against artifacts associated with changes in labeling density (*Coltharp et al., 2014*). Consequently, our data indicates the movement of proteins between domains remains largely independent of potential artifacts stemming from artificial clustering (*Moon et al., 2017*; *Petersen et al., 2020*; *Robinson et al., 2019*). This conclusion is further supported by the consistency of dSTORM results obtained using both EGFP and cy3b-anti-TREK-1 antibody (*Figure 2*, *Figure 2—figure supplement 2*).

# Materials and methods

**Key resources table**

| Reagent type (species) or resource | Designation | Source or reference | Identifiers | Additional information |
|---|---|---|---|---|
| Cell line (*Homo sapiens*) | HEK293T | ATCC | HEK293T | Cat# CRL-3216; RRID:CVCL_0063 |
| Cell line (*Mus musculus*) | N2a | ATCC | N2a | Cat# CCL-131; RRID:CVCL_0470 |
| Cell line (*M. musculus*) | C2C12 | ATCC | C2C12 | Cat# CRL-1772 |
| Transfected construct (*H. sapiens*) | TREK-1 | PMID:18004376 | Dr. Steven Long (Sloan Kettering) | |
| Transfected construct (*M. musculus*) | mPLD2 | PMID:9867870 | Dr. Michael Frohman (Stony Brook) | |
| Transfected construct (*M. musculus*) | xPLD2 | PMID:9867870 | Dr. Michael Frohman (Stony Brook) | Catalytically dead PLD2 |
| Cell line (*Drosophila melanogaster*) | BG2-c2 | Drosophila Genomics Resource Center | DGRC Stock 53; https://dgrc.bio.indiana.edu//stock/53; RRID:CVCL_Z719 | |
| Strain, strain background (*Drosophila melanogaster*) | PLDnull | PMID:27848911 | | |
| strain, strain background (*D. melanogaster*) | PLD-KD | Vienna *Drosophila* Resource Center | Stock#: v106137 | |
| Biological sample (*M. musculus*) | SREBP2-KO | PMID:34385305 | Heather Ferris (University of Virginia) | Brain slices from SREBP2-KO animals |
| Antibody | Anti-TREK-1 (rabbit, polyclonal) | Santa Cruz | Cat# sc-50412; RRID:AB_2131048; | 1:100 dilution |
| Antibody | Anti-TREK-1 (mouse, monoclonal) | Santa Cruz | Cat# sc-398449 | 1:100 dilution |
| Antibody | Anti-PIP2 (mouse, monoclonal) | Echelon Biosciences | Cat# Z-P045, RRID:AB_427225 | 1:100 dilution |
| Antibody | Anti-rabbit Alexa 647 (goat, polyclonal) | Thermo Fisher Scientific | Cat# A-21244, RRID:AB_2535812 | 1:1000 dilution |
| Antibody | Anti-mouse Alexa 647 (goat, polyclonal) | Thermo Fisher Scientific | Cat# A-21235, RRID:AB_2535804 | 1:1000 dilution |
| Antibody | Anti-mouse cy3B (donkey, polyclonal) | PMID:27976674 | Cat# NC9812063 | 1:1000 dilution |
| Chemical compound, drug | CTxB | Thermo Fisher Scientific | Cat# C34778 | |
| Peptide, recombinant protein | ApoE3 | BioLegend, USA | Cat# 786802 | |
| Chemical compound, drug | Atto 647 | Sigma-Aldrich | 18373-1MG-F | |

*Continued on next page*

*Continued*

| Reagent type (species) or resource | Designation | Source or reference | Identifiers | Additional information |
|---|---|---|---|---|
| Chemical compound, drug | Cholesterol oxidase | Sigma-Aldrich | C8649-250UN | |
| Chemical compound, drug | Amplex red | Cayman Chemical | Cat# 10010469 | |
| Peptide, recombinant protein | Horseradish peroxidase | VWR | 516531-5KU | |
| Peptide, recombinant protein | Choline oxidase | VWR | Cat# 15349250 | |
| Chemical compound, drug | C8-PC | Avanti Lipids | Cat# 850315P | |
| Peptide, recombinant protein | Glucose oxidase | Sigma-Aldrich | Cat# G2133 | |
| Peptide, recombinant protein | Catalase | Sigma-Aldrich | Cat# C40 | |
| Chemical compound, drug | Maleimide cy3B | GE-Health | Cat# PA63131 | |

## Cell culture and gene expression

HEK293T cells (ATCC Cat# CRL-3216, RRID:CVCL_0063), C2C12 cells (ATCC Cat# CRL-1772), and N2a (ATCC Cat# CCL-131) were cultured in DMEM (Corning cellgro) with 10% FBS, 100 units/ml penicillin, and 100 µg/ml streptomycin. Mycoplasma testing was performed by PCR and found to be negative. Cells were plated on poly-D-lysine-coated 12 mm microscope cover glass at approximately 12 hr, 36 hr, or 60 hr before transfection with genes for target proteins. Transfections were performed using X-tremeGENE 9 DNA transfection agent (Roche Diagnostics). Full-length human TREK-1(TREK-1 FL) with C-terminus GFP tag in pCEH vector was obtained from Dr. Stephen Long. Mouse PLD2 constructs (mPLD2) and inactive mutant (K758R single mutation, xPLD2) without GFP tag in pCGN vector were provided by Dr. Michael Frohman. TREK-1 constructs were co-transfected with mPLD2 or xPLD2 at a 1:4 ratio (0.5 g of TREK-1 and 2 g of PLD DNA) (*Comoglio et al., 2014*). All the salts for internal/external solutions were purchased from either Sigma or Fisher Scientific.

## Electrophysiology

The transfected HEK293T cells were used in 24–36 hr after transfection for standard excised inside-out patch-clamp recordings of TREK-1 (*Brohawn et al., 2014a*; *Comoglio et al., 2014*). Currents were recorded blinded at room temperature using an Axopatch 200B amplifier and Digidata 1440A (Molecular Devices). Borosilicate glass electrode pipettes (B150-86-10, Sutter Instrument) were pulled with the Flaming/Brown micropipette puller (Model P-1000, Sutter Instrument), resulting in 3–6 MΩ resistances with the pipette solution (in mM): 140 NaCl, 5 KCl, 1 CaCl$_2$, 3 MgCl$_2$, 10 TEA-Cl, 10 HEPES, pH 7.4 (adjusted with NaOH). Bath solution consists of (in mM) 140 KCl, 3 MgCl$_2$, 5 EGTA, 10 TEA-Cl, 10 HEPES, pH 7.2 (adjusted with KOH). Low concentration of TEA (10 mM) was added into both pipette/bath solutions to block the endogenous potassium channels in HEK293T cells. Patch electrodes were wrapped with parafilm to reduce capacitance. Currents measured using Clampex 10.3 (Molecular Devices) were filtered at 1 kHz, sampled at 20 kHz, and stored on a hard disk for later analysis.

Pressure clamping on the patch was performed using high-speed pressure clamping system (ALA Scientific) through the Clampex control. Data was analyzed offline by a homemade procedure using IgorPro 6.34A (WaveMetrics).

TREK-1 current, either co-expressed with mPLD2 or xPLD2, was activated by negative pressure steps from 60 to 0 mmHg in 10 mmHg decrements at +30 mV membrane potential, and five traces for each case were recorded and averaged for the analysis. Patch size was estimated using the current density (I_density; pA/µm$^2$). Then, a Boltzmann equation, I_density = base + {max/[1+exp((P50-P)/slope)]} was used to fit the data with a constraint of base = 1 due to poor saturation of the current at high pressure. P is the applied pressure, P50 is the pressure that activates 50% of maximum current measured (non-saturating), and slope shows the sensitivity of current activation by pressure. In some experiments with hTREK-1+xPLD2 co-expression where the activated currents were too small to fit to the Boltzmann equation, the current amplitude at P=-30 mmHg (I_m30) was compared with its 5× standard deviation(I_5×SD). If I_m30 < I_5×SD, the experiment was excluded from the Boltzmann equation fitting and corresponding P50-slope analysis. This empirical rule (we call it 5×SD rule)

can discern four out of five wild-type (TREK-1 FL) cell-attached recording cases as null experiments, suggesting that it could be a usable/useful empirical criterion for our experiment. Then, the current density at –60 mmHg and P50-slope data were used for statistical analysis. Mann–Whitney test was done to assess statistical significance using Prism6 (GraphPad Software), and outliers were eliminated using a built-in function in Prism with Q = 1%. The values represented are mean ± SEM.

## Animals

Housing, animal care, and experimental procedures were consistent with the Guide for the Care and Use of Laboratory Animals and approved by the Institutional Animal Care and Use Committee of the University of Virginia. The tissue from SREBP2 mice are from strains 030826 and 004600 from the Jackson Laboratories as described in *Wang et al., 2021*. No new animals were used for this study.

## Fixed cell preparation

Cells (C2C12, HEK293T, and N2a) were grown to 80% confluence (C2C12 were allowed to differentiate overnight in serum free media). Cells were rinsed, treated as needed, and then fixed with 3% paraformaldehyde and 0.1% glutaraldehyde for 10 min to fix both protein and lipids. Glutaraldehyde was reduced with 0.1% $NaBH_4$ for 7 min followed by three 10 min washes with PBS. Cells were permeabilized for 15 min with 0.2% Triton X-100, blocked with 10% BSA/0.05% Triton/PBS at room temperature for 90 min. Primary antibody (PLD2, Cell Signaling #13891; TREK-1, Santa Cruz #sc-50412; $PIP_2$, Echelon Biosciences #z-P045) was added to a solution of 5% BSA/0.05% Triton/PBS for 60 min at rt at a concentration of 1:100 followed by five washes with 1% BSA/0.05% Triton/PBS for 15 min each. Secondary antibody (Life Technologies #A21244 and A21235; cy3B antibodies were produced as described previously; *Petersen et al., 2016*) was added in the same buffer as primary for 30 min at room temperature followed by five washes as above. A single 5 min wash with PBS was followed by a post-fix with fixing mixture, as above, for 10 min w/o shaking. This was followed by three 5 min washes with PBS and two 3 min washes with $dH_2O$. Cells only receiving CTxB treatment were not permeabilized.

Brain slices from a hGFAP-Cre driving the SB2 knockout mouse and its Flox control were from the same animals that were previously characterized (*Wang et al., 2021*). The protocol is the same as cells except the permeabilized samples were treated with anti-TREK cy3b+CTxB-647 for 3 d (4°C) and then washed five times (room temperature) for 1 hr prior to post fixing.

The dual-fixation protocol is used to minimize any effects from post-fixation aberrations. While always good practice for super-resolution in general, this dual fixation also ensures that the movement of any molecule of interest which may not have been immobilized by the initial fixation can be fully immobilized after labeling since the antibodies or toxins used for labeling will be efficiently cross-linked during this post-labeling fixation step. While some have proposed that this problem should be solved by adding the label before the initial fixation (*Tanaka et al., 2010*), we believe that in the absence of easily attainable monomeric labeling molecules it would have likely led to clustering artifacts due to the (often) multimeric nature of the labeling proteins.

For cells loaded with cholesterol, 4 ug/ml apolipoprotein E3 (BioLegend, USA) was mixed with fresh 10% FBS and applied to the cells for 1 hr prior to shear and/or fixing.

Shear force was applied to cells in ibidi μ-Slide $I^{0.4}$ Luer chambers with a flow rate calibrated to apply 3.0 dynes/cm$^2$. Fixation media (see above) was applied to cells using a syringe pump (Harvard Apparatus PHD ULTRA) and kept at 37°C using an in-line heater (Warner SH-27B).

TREK-1 in HEK293T cells was labeled with a EGFP concatenated to the C-Terminus of full-length human TREK-1 (*Figure 1—figure supplement 2B*) or by applying anti-TREK-1 antibody (sc-398449, Santa Cruz) conjugated to cy3b (*Petersen et al., 2016*). Anti-$PIP_2$ antibody was directly conjugate Alexa 647 using the same protocol.

## Super-resolution dSTORM

Images were recorded with a Vutara 352 and VXL super-resolution microscopes (Bruker Nano Surfaces, Salt Lake City, UT), which is based on the 3D Biplane approach. Super-resolution images were captured using a Hamamatsu ORCA Flash4.0 sCMOS camera and a ×60 water objective with numerical aperture 1.2. Biological replicates (6–12) are individual cells from at least two independent experiments. Data were analyzed using the Vutara SRX software (version 5.21.13 for the data

in *Figures 3–5* and version 7.0.07 for the data in *Figure 2*). Single molecules were identified by their brightness frame by frame after removing the background. Identified particles were then localized in three dimensions by fitting the raw data in a customizable region of interest (typically 16 × 16 pixels) centered on each particle in each plane with a 3D model function that was obtained from recorded bead datasets (10,000–750,000 localization per biological replicate). Fit results were stored as data lists for further analysis.

Fixed samples were imaged using a 647 nm and 561 nm excitation lasers, respectively, and 405 nm activation laser in photo switching buffer comprising of 20 mM cysteamine (Sigma, #30070), 1% beta-mercaptoethanol (BME) (Sigma, #63689), and oxygen scavengers (glucose oxidase, GLOX) (Sigma #G2133) and catalase (Sigma #C40) in 50 mM Tris (Affymetrix, #22638100) + 10 mM NaCl (Sigma, #S7653) + 10% glucose (Sigma, #G8270) at pH 8.0 at 50 Hz and maximal powers of 647 nm, 561 nm and 405 lasers set to 8, 10, and 0.05 kW cm$^{-2}$, respectively.

## Three-color EGFP-STORM

The dSTORM with EGFP (EGFP-STORM) was performed identical to the two-color dSTORM described above except that a 488 laser was also used to excite EGFP and PIP$_2$ and CTxB were directly conjugated with fluorescent dyes (Atto 647 and cy3b 555, respectively), obviating the need for fluorescent secondary antibodies. TREK-1 and PIP$_2$ antibodies were conjugated with NHS esters of either Cy3B or Atto 647. 1.5 mg of antibody was conjugated to 3 ng of dye in 1 M NaHCO$_3$ buffer pH 8 for 1 hr at room temperature and separate on a NAP-5 desalting column. The acquisition was performed with no 405-activation laser in GLOX/BME buffer. The GLOX/BME buffer was not required for EGFP blinking, but it did improve the fluorescence and the number of localization particles determined. The resolution TREK-1 localizations determined by EGFP_TREK-1 and cy3b-labeled anti TREK-1 antibody were comparable (45–50 nm resolution).

The pair correlation function g(r) and cluster analysis were performed using the Statistical Analysis package in the Vutara SRX software. Pair correlation analysis is a statistical method used to determine the strength of correlation between two objects by counting the number of points of probe 2 within a certain donut-radius of each point of probe 1. This allows for localization to be determined without overlapping pixels as done in traditional diffraction-limited microscopy. For three-color EGFP-STORM, probes 1 and 3 and 2 and 3 were also compared using the pair correlation function. Localization at super resolution is beyond techniques appropriate for diffraction-limited microscopy such as Pearson's correlation coefficient. Lipid cluster size was determined using the DBSCAN clustering algorithm also included as part of the Vutara SRX software.

## Fluorescence recovery after photobleaching (FRAP)

C2C12 cells were grown in DMEM with 10% FBS until 16 hr before use in which they were switched into serum-free DMEM. On the day of the experiment, DMEM in live cells was replaced with DMEM w/o phenol red. Fixed cells were rinsed once with PBS and then put into a mixture of PBS with 3% PFA and 0.1% glutaraldehyde for 20 min at 37°C. Fixed cells were then rinsed with PBS 5 × 5 min and placed back into phenol-free DMEM. CTxB (Thermo Fisher C34778, 100 ug/ml) was then applied 1:200 into each plate and allowed to incubate for >30 min before imaging. Imaging and data collection was performed on a Leica SP8 confocal microscope with the Application Suite X v.1.1.0.12420. Five images were taken as baseline after which a selection of one or more ROI was bleached at 100% laser power for 6–8 frames. Recovery was measured out to 5 min, and fluorescence of the ROI(s) was quantified. The fluorescence before bleaching was normalized to 1 and after the bleaching step was normalized to 0.

## Cholesterol assay

N2a cells were cultured in 48-well plates with 200 ul media in each well and then changed to 200 ul PBS for the shear treatment. The shear plate was incubated with PBS on an orbital rotator at 3 dynes/cm$^2$ for 10 min in a 37°C incubator. The control plate was incubated with PBS for 10 min in the same incubator with no shear. Then the shear plate was incubated with 200 ul 4%PFA + 0.1% glutaraldehyde in PBS for 10 min with 3 dynes/cm$^2$ shear and 10 min without shear. The control plate was fixed for 20 min with no shear. Fixed cells were lysed in RIPA buffer (Thermo) and assay in 100 µl PBS with, 4 U/ml cholesterol oxidase, 100 µM amplex red, and 2 U/ml horseradish peroxidase (HRP). The data

shown in the figure are from two biological replicates (independent experiments) with 5–10 technical replicates.

## In vitro cellular PLD assay

In vitro cellular PLD2 activity was measured in cultured HEK293T cells by an enzyme-coupled product release assay (*Petersen et al., 2016*) using amplex red reagent. Cells were seeded into 96-well plates (~5 × 10⁴ cells per well) and incubated at 37°C overnight to reach confluency. The cells were starved with serum-free DMEM for a day and washed once with PBS. The PLD reaction was initiated by adding 100 µl of reaction buffer (100 µM amplex red, 2 U/ml HRP, 0.2 U/ml choline oxidase, and 60 µM C8-PC in PBS). The assay reaction was performed for 2–4 hr at 37°C, and the activity was kinetically measured with a fluorescence microplate reader (Tecan Infinite 200 Pro) at excitation and emission wavelengths of 530 nm and 585 nm, respectively. The PLD2 activity was calculated by subtracting the background activity (reaction buffer, but no cells). For the bar graphs, samples were normalized to the control activity at the 120 min time point.

## *Drosophila* assays

For behavior experiments, 1- to 5-day-old flies were collected in vials containing ~50 flies at least 12 hr before the experiment. Flies were allowed to acclimate to behavior room conditions for >30 min (dim red light, ~75% humidity) before each assay. Shock avoidance was tested by placing flies in a T-maze where they could choose between an arm shocking at the indicated voltage every 2 s and an arm without shock. Flies were given 2 min to choose which arm, after which flies were collected and counted to determine the shock avoidance index for each voltage and genotype. Control and knockout flies were alternated to avoid any preference, and the arm used for shock was also alternated to control for any non-shock preference in the T-maze itself. Shock avoidance index (AI) was calculated as AI = (# flies shock arm-# flies control arm)/# flies total. Plotting the inverse of this metric, we obtain a pain sensitivity curve in which we observe a right-shift when the *pld* gene was knocked down (*Figure 3F*). Flies were averaged from 2 to 3 biological replicates with ~10 groups per replicate.

Arousal threshold protocol has been described in detail previously (*Murphy et al., 2016*). Briefly, animals were exposed hourly to a series of vibrations of increasing intensity ranging from 0.8 to 3.2 g, in steps of 0.6 g. Stimuli trains were composed of 200 ms vibration with 800 ms inter-vibration interval and 15 s inter-stimuli train interval. Stimulation intensity and timing were controlled using pulse-width modulation via an Arduino UNO and shaftless vibrating motors (Precision Microdrives, model 312-110). Arousal to a given stimulus was assigned when an animal (1) was inactive at the time of the stimulus, (2) satisfied a given inactivity criteria at the time of the stimulus, and (3) moved within the inter-stimuli train period (15 s) of that stimulus. Statistics are from two biological replicates.

## Statistics

All statistical calculations were performed using a Student's *t*-test or Mann–Whitney test in Prism software (v9) unless otherwise noted. For statistics of more than one point along a pair correlation curve, a nested Student's *t*-test was used. Significance is noted as follows: ns, $p > 0.05$; *$p < 0.05$; **$p < 0.01$; ***$p < 0.001$; ****$p < 0.0001$.

## Acknowledgements

We thank Tamara Boto and Seth Tomchik for their assistance in the *Drosophila* shock experiments, Michael Frohman from Stony Brook for the mouse PLD and mutant PLD cDNA, Steven Long from Memorial Sloan Kettering for human TREK-1-GFP, Padinjat Raghu for the PLD mutant *Drosophila*, Andrew S Hansen for PLD experiments, multiple aspects of experimental design and discussion, Yul Young Park for the electrophysiology experimentation, and Carl Ebeling for his help and discussion on the imaging analysis. This work was supported by a Director's New Innovator Award to SBH (DP2NS087943), R21 (AG078845-01), and R01 (R01NS112534) from the National Institutes of Health, an R01 to WWJ (R01AG045036) from the National Institute on Aging, and a graduate fellowship from the Joseph B Scheller & Rita P Scheller Charitable Foundation to ENP. We are grateful to the JPB Foundation for the purchase of a super-resolution microscope. The authors declare no conflict of interest.

## Additional information

### Funding

| Funder | Grant reference number | Author |
|---|---|---|
| National Institutes of Health | DP2NS087943-01 | Scott B Hansen |
| National Institutes of Health | R01NS112534 | Scott B Hansen |
| National Institutes of Health | R01AG045036 | William Ja |

The funders had no role in study design, data collection and interpretation, or the decision to submit the work for publication.

### Author contributions

E Nicholas Petersen, Conceptualization, Data curation, Formal analysis, Investigation, Methodology, Writing – original draft; Mahmud Arif Pavel, Data curation, Formal analysis, Investigation, Methodology, Writing – original draft; Samuel S Hansen, Zixuan Yuan, Investigation; Manasa Gudheti, Investigation, Methodology; Hao Wang, Investigation, Methodology, Writing – review and editing; Keith R Murphy, Data curation, Investigation, Methodology; William Ja, Resources, Supervision, Writing – review and editing; Heather A Ferris, Erik Jorgensen, Resources, Writing – review and editing; Scott B Hansen, Conceptualization, Resources, Data curation, Formal analysis, Supervision, Funding acquisition, Validation, Investigation, Methodology, Writing – original draft, Project administration, Writing – review and editing

### Author ORCIDs

E Nicholas Petersen ⓘ http://orcid.org/0000-0002-3243-9611
Zixuan Yuan ⓘ http://orcid.org/0000-0003-0665-5086
Scott B Hansen ⓘ https://orcid.org/0000-0003-0086-9753

Reviewer #1 (Public Review): https://doi.org/10.7554/eLife.89465.3.sa1
Reviewer #2 (Public Review): https://doi.org/10.7554/eLife.89465.3.sa2
Reviewer #3 (Public Review): https://doi.org/10.7554/eLife.89465.3.sa3
Author Response https://doi.org/10.7554/eLife.89465.3.sa4

## Additional files

### Supplementary files

• MDAR checklist

### Data availability

Electrophysiology, dSTORM (pair correlation and cluster analysis) for shear and osmotic shock, cholesterol and PLD assays, and fly behavior data are available at Mendeley Data, V1, https://doi.org/10.17632/pbj4nx55jt.1.

The following dataset was generated:

| Author(s) | Year | Dataset title | Dataset URL | Database and Identifier |
|---|---|---|---|---|
| Nicholas PE, Hansen SB | 2024 | Membrane mediated TREK mechanosensation | https://doi.org/10.17632/pbj4nx55jt.1 | Mendeley Data, 10.17632/pbj4nx55jt.1 |

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
