## [Editor Report · eLife assessment]

This **important** study poses a provocative mechanism of channel activation of the mechanically activated ion channel TREK-1. The data provide **solid** evidence that the application of shear to cells causes a redistribution of both TREK-1 and an associated enzyme, PhospholipaseD2 in the membrane that increases the enzyme activity. The work offers a new mechanism, but note that this is only one possible method of channel activation, and mechanisms independent of PLD2 are also probable.

---

## [Referee Report · Reviewer #1 (Public Review)]

Force sensing and gating mechanisms of the mechanically activated ion channels is an area of broad interest in the field of mechanotransduction. These channels perform important biological functions by converting mechanical force into electrical signals. To understand their underlying physiological processes, it is important to determine gating mechanisms, especially those mediated by lipids. The authors in this manuscript describe a mechanism for mechanically induced activation of TREK-1 (TWIK-related K+ channel). They propose that force induced disruption of ganglioside (GM1) and cholesterol causes relocation of TREK-1 associated with phospholipase D2 (PLD2) to 4,5-bisphosphate (PIP2) clusters, where PLD2 catalytic activity produces phosphatidic acid that can activate the channel. To test their hypothesis, they use dSTORM to measure TREK-1 and PLD2 colocalization with either GM1 or PIP2. They find that shear stress decreases TREK-1/PLD2 colocalization with GM1 and relocates to cluster with PIP2. These movements are affected by TREK-1 C-terminal or PLD2 mutations suggesting that the interaction is important for channel re-location. The authors then draw a correlation to cholesterol suggesting that TREK-1 movement is cholesterol dependent. It is important to note that this is not the only method of channel activation and that one not involving PLD2 also exists. Overall, the authors conclude that force is sensed by ordered lipids and PLD2 associates with TREK-1 to selectively gate the channel.

The proposed mechanism is solid and the authors have revised the manuscript to address previous issues with the first version

---

## [Referee Report · Reviewer #2 (Public Review)]

This manuscript by Petersen and colleagues investigates the mechanistic underpinnings of activation of the ion channel TREK-1 by mechanical inputs (fluid shear or membrane stretch) applied to cells. Using a combination of super-resolution microscopy, pair correlation analysis and electrophysiology, the authors convincingly show that the application of shear to a cell can lead to changes in the distribution of TREK-1 and the enzyme PhospholipaseD2 (PLD2), relative to lipid domains defined by either GM1 or PIP2. The activation of TREK-1 by mechanical stimuli was shown to be sensitized by the presence of PLD2, but not a catalytically inactive xPLD2 mutant. In addition, the activity of PLD2 is increased when the molecule is more associated with PIP2, rather than GM1 defined lipid domains. The presented data do not exclude direct mechanical activation of TREK-1, rather suggest a modulation of TREK-1 activity, increasing sensitivity to mechanical inputs, through an inherent mechanosensitivity of PLD2 activity. The authors additionally demonstrate that cellular uptake of cholesterol inhibits TREK-1 activation and, in ex vivo studies, that depletion of cholesterol from astrocytes reduces correlation of TREK-1 and G1 lipids in mouse brain slices. In vivo studies, using *Drosophila melanogaster* behavioural assays, were used to demonstrate that disrupting PLD2 altered behavioural responses to mechanical and electrical inputs. These data demonstrate that manipulation of PLD2 analogue in the fly can alter sensory transduction, suggesting that PLD functions to regulate sensitivity to mechanical force. However, as the authors note, there is no TREK-1 homologue in this organism: thus the identity of the downstream effectors of PLD in *D. melanogaster* remain unknown. This work will be of interest to the growing community of scientists investigating the myriad mechanisms that can tune mechanical sensitivity of cells, providing valuable insight into the role of functional PLD2 in sensitizing TREK-1 activation in response to mechanical inputs, in some cellular systems.

The authors convincingly demonstrate that, post application of shear, an alteration in the distribution of TREK-1 and mPLD2 (in HEK293T cells) from being correlated with GM1 defined domains (no shear) to increased correlation with PIP2 defined membrane domains (post shear). The association of TREK-1 with PIP2 required functional mPLD2. These data were generated using super-resolution microscopy to visualise, at sub diffraction resolution, the localisation of labelled protein, compared to labelled lipids. The use of super-resolution imaging enabled the authors to visualise changes in cluster association that would not have been achievable with diffraction limited microscopy.

This work provides further evidence of the astounding flexibility of mechanical sensing in cells. By outlining how mechanical activation of TREK-1 can be sensitised by mechanical regulation of PLD2 activity, the authors highlight a mechanism by which TREK-1 sensitivity could be regulated under distinct physiological conditions.

---

## [Referee Report · Reviewer #3 (Public Review)]

The manuscript "Mechanical activation of TWIK-related potassium channel by nanoscopic movement and second messenger signaling" presents a new mechanism for the activation of TREK-1 channel. The mechanism suggests that TREK1 is activated by phosphatidic acids that are produced via a mechanosensitive motion of PLD2 to PIP2-enriched domains. Overall, I found the topic interesting but several typos and unclarities reduced the readability of the manuscript. Additionally, I have several major concerns on the interpretation of the results. Therefore, the proposed mechanism is not fully supported by the presented data. Lastly, the mechanism is based on several previous studies from the Hansen lab, however, the novelty of the current manuscript is not clearly stated. For example, in the 2nd result section, the authors stated, "fluid shear causes PLD2 to move from cholesterol dependent GM1 clusters to PIP2 clusters and this activated the enzyme". However, this is also presented as a new finding in section 3 "Mechanism of PLD2 activation by shear."

In the revised manuscript, the authors addressed most of my concerns. I still have the following suggestions/confusions.

1. the reviewer would highly appreciate verification of the cholesterol assay, either by additional experiment or by citations of independent work.

2. The claim on "shear thinning" is still very confusing. First, asymmetric insertion of molecules to one monolayer of the membrane is a main mechanism for membrane bending and curvature formation. Second, why is "shear thinning" equivalent to entropy/order?

---

## [Author Response]

The following is the authors’ response to the original reviews.

We sincerely thank the reviewers for their in-depth consideration of our manuscript and their helpful reviews. Their efforts have made the paper much better. We have responded to each point. The previously provided public responses have been updated they are included after the private response for convenience.

**Reviewer #1 (Recommendations For The Authors):**
1. In general, the manuscript will benefit from copy editing and proof reading. Some obvious edits;Page 6 line 140. Do the authors mean Cholera toxin B?

Response: We corrected this error and went through the entire paper carefully correcting for grammar and increased clarity.

Page 8 line 173. Methylbetacyclodextrin is misspelled.

Response: Yes, corrected.

Figure 4c is missing representative traces for electrophysiology data.Figure 4. Please check labeling ordering in figure legend as it does not match the panels in the figure.

Thank you for the correction and we apologize for the confusion in figure 4. We uploaded an incomplete figure legend, and the old panel ‘e’ was not from an experiment that was still in the figure. It was removed and the figure legends are now corrected.

Please mention the statistical analysis used in all figure legends.

Response: Thank you for pointing out this omission, statistics have been added.

Although the schematics in each figure helps guide readers, they are very inconsistent and sometimes confusing. For example, in Figure 5 the gating model is far-reaching without conclusive evidence, whereas in Figure 6 it is over simplified and unclear what the image is truly representing (granted that the downstream signaling mechanism and channel is not known).

Response: Figure 5d is the summary figure for the entire paper. We have made this clearer in the figure legend and we deleted the title above the figure that gave the appearance that the panel relates to swell only. It is the proposed model based on what we show in the paper and what is known about the activation mechanism of TREK-1.

Figure 6 is supposed to be simple. It is to help the reader understand that when PA is low mechanical sensitivity is high. Without the graphic, previous reviewers got confused about threshold going down and mechanosensitivity going up and how the levels of PA relate. Low PA = high sensitivity. We’ve added a downstream effector to the right side of the panel to avoid any biased to a putative downstream channel effector. The purpose of the experiment is to show PLD has a mechanosensitive phenotype in vivo.

**Reviewer #2 (Recommendations For The Authors):**
This manuscript outlines some really interesting findings demonstrating a mechanism by which mechanically driven alterations in molecular distributions can influence (a) the activity of the PLD2 molecule and subsequently (b) the activation of TREK-1 when mechanical inputs are applied to a cell or cell membrane.The results presented here suggest that this redistribution of molecules represents a modulatory mechanism that alters either the amplitude or the sensitivity of TREK-1 mediated currents evoked by membrane stretch. While the authors do present values for the pressure required to activate 50% of channels (P50), the data presented provides incomplete evidence to conclude a shift in threshold of the currents, given that many of the current traces provided in the supplemental material do not saturate within the stimulus range, thus limiting the application of a Boltzmann fit to determine the P50. I suggest adding additional context to enable readers to better assess the limitations of this use of the Boltzmann fit to generate a P50, or alternately repeating the experiments to apply stimuli up to lytic pressures to saturate the mechanically evoked currents, enabling use of the Boltzmann function to fit the data.

Response: We thank the reviewer for pointing this out. We agree the currents did not reach saturation. Hence the term P50 could be misleading, so we have removed it from the paper. We now say “half maximal” current measured from non-saturating pressures of 0-60 mmHg. We also deleted the xPLD data in supplemental figure 3C since there is insufficient current to realistically estimate a half maximal response.

In my opinion, the conclusions presented in this manuscript would be strengthened by an assessment of the amount of TREK-1 in the plasma membrane pre and post application of shear. While the authors do present imaging data in the supplementary materials, these data are insufficiently precise to comment on expression levels in the membrane. To strengthen this conclusion the authors could conduct cell surface biotinylation assays, as a more sensitive and quantitative measure of membrane localisation of the proteins of interest.

1. Response: as mentioned previously, we do not have an antibody to the extracellular domain. Nonetheless to better address this concern we directly compared the levels of TREK-1, PIP2, and GM1; in xPLD2, mPLD2, enPLD2 with and without shear. The results are in supplemental figure 2. PLD2 is known to increase endocytosis1 and xPLD2 is known to block both agonist induced and constitutive endocytosis of µ-opioid receptor2. The receptor is trapped on the surface. This is true of many proteins including Rho3, ARF4, and ACE21 among others. In agreement with this mechanism, in Figure S2C,G we show that TREK increases with xPLD and the localization can clearly be seen at the plasma membrane just like in all of the other publications with xPLD overexpression. xPLD2 would be expected to inhibit the basal current but we presume the increased expression likely has compensated and there is sufficient PA and PG from other sources to allow for the basal current. It is in this state that we then conduct our ephys and monitor with a millisecond time resolution and see no activation. We are deriving conclusion from a very clear response—Figure 1b shows almost no current, even at 1-10 ms after applying pressure. There is little pressure current when we know the channel is present and capable of conducting ion (Figure 1d red bar).After shear there is a strong decrease in TREK-1 currents on the membrane in the presence of xPLD2. But it is not less than TREK-1 expression with mPLD2. And since mouse PLD2 has the highest basal current and pressure activation current. The amount of TREK-1 present is sufficient to conduct large current. To have almost no detective current would require at least a 10 fold reduction compared to mPLD2 levels before we would lack the sensitivity to see a channel open. Lasty endocytosis typically in on the order of seconds to minutes, no milliseconds.

2. We have shown an addition 2 independent ways that TREK-1 is on the membrane during our stretch experiments. Figure 1d shows the current immediately prior to applying pressure for wt TREK-1. When catalytically dead PLD is present (xPLD2) there is almost normal basal current. The channel is clearly present. And then in figure 1a we show within a millisecond there is no pressure current. As a control we added a functionally dead TREK-1 truncation (xTREK). Compared to xPLD2 there is clearly normal basal current. If this is not strong evidence the channel was available on the surface for mechanical activation please help us understand why. And if you think within 2.1 ms 100% of the channel is gone by endocytosis please provide some evidence that this is possible so we can reconsider.

3. We have TIRF super resolution imaging with ~20 nm x-y resolution and ~ 100nm z resolution and Figure 2b clearly shows the channel on the membrane. When we apply pressure in 1b, the channel is present.

4. Lastly, In our previous studies we showed activation of PLD2 by anesthetics was responsible for all of TREK-1’s anesthetic sensitivity and this was through PLD2 binding to the C-terminus of TREK-15. We showed this was the case by transferring anesthetic sensitivity to an anesthetic insensitive homolog TRAAK. This established conclusively the basic premise of our mechanism. Here we show the same C-terminal region and PLD2 are responsible for the mechanical current observed by TREK-1. TRAAK is already mechanosensitive so the same chimera will not work for our purposes here. But anesthetic activation and mechanical activation are dramatically different stimuli, and the fact that the role of PLD is robustly observed in both should be considered.

The authors discuss that the endogenous levels of TREK-1 and PLD2 are "well correlated: in C2C12 cells, that TREK-1 displayed little pair correlation with GM1 and that a "small amount of TREK-1 trafficked to PIP2". As such, these data suggest that the data outlined for HEK293T cells may be hampered by artefacts arising from overexpression. Can TREK-1 currents be activated by membrane stretch in these cells C2C12 cells and are they negatively impacted by the presence of xPLD2? Answering this question would provide more insight into the proposed mechanism of action of PLD2 outlined by the authors in this manuscript. If no differences are noted, the model would be called into question. It could be that there are additional cell-specific factors that further regulate this process.

Response: The low pair correlation of TREK-1 and GM1 in C2C12 cells was due to insufficient levels of cholesterol in the cell membrane to allow for robust domain formation. In Figure 4b we loaded C2C12 cells with cholesterol using the endogenous cholesterol transport protein apoE and serum (an endogenous source of cholesterol). As can be seen in Fig. 4b, the pair correlation dramatically increased (purple line). This was also true in neuronal cells (N2a) (Fig 4d, purple bar). And shear (3 dynes/cm2) caused the TREK-1 that was in the GM1 domains to leave (red bar) reversing the effect of high cholesterol. This demonstrates our proposed mechanism is working as we expect with endogenously expressed proteins.

There are many channels in C2C12 cells, it would be difficult to isolate TREK-1 currents, which is why we replicated the entire system (ephys and dSTORM) in HEK cells. Note, in figure 4c we also show that adding cholesterol inhibits TREK-1 whole cell currents in HEK293cells.

As mentioned in the public review, the behavioural experiments in *D. melanogaster* can not solely be attributed to a change in threshold. While there may be a change in the threshold to drive a different behaviour, the writing is insufficiently precise to make clear that conclusions cannot be drawn from these experiments regarding the functional underpinnings of this outcome. Are there changes in resting membrane potential in the mutant flys? Alterations in Nav activity? Without controlling for these alternate explanations it is difficult to see what this last piece of data adds to the manuscript, particularly given the lack of TREK-1 in this organism. At the very least, some editing of the text to more clearly indicate that these data can only be used to draw conclusions on the change in threshold for driving the behaviour not the change in threshold of the actual mechanotransduction event (i.e. conversion of the mechanical stimulus into an electrochemical signal).

Response: We agree; features other than PLDs direct mechanosensitivity are likely contributing. This was shown in figure 6g left side. We have an arrow going to ion channel and to other downstream effectors. We’ve added the putative alteration to downstream effectors to the right side of the panel. This should make it clear that we no more speculate the involvement of a channel than any of the other many potential downstream effectors. As mentioned above, the figure helps the reader coordinate low PA with increased mechanosensitivity. Without the graphic reviewers got confused that PA increased the threshold which corresponds to a decreased sensitivity to pain. Nonetheless we removed our conclusion about fly thresholds from the abstract and made clearer in the main text the lack of mechanism downstream of PLD in flies including endocytosis. Supplemental Figure S2H also helps emphasize this. .

Nav channels are interesting, and since PLD contribute to endocytosis and Nav channels are also regulated by endocytosis there is likely a PLD specific effect using Nav channels. There are many ways PA likely regulates mechanosensitive thresholds, but we feel Nav is beyond the scope of our paper. Someone else will need to do those studies. We have amended a paragraph in the conclusion which clearly states we do not know the specific mechanism at work here with the suggestions for future research to discover the role of lipid and lipid-modifying enzymes in mechanosensitive neurons.

There may be fundamental flaws in how the statistics have been conducted. The methods section indicates that all statistical testing was performed with a Student's t-test. A visual scan of many of the data sets in the figures suggests that they are not normally distributed, thus a parametric test such as a Student's t-test is not valid. The authors should assess if each data set is normally distributed, and if not, a non-parametric statistical test should be applied. I recommend assessing the robustness of the statistical analyses and adjusting as necessary.

Response: We thank the reviewer for pointing this out, indeed there is some asymmetry in Figure 6C-d. The p values with Mann Whitney were slightly improved p=0.016 and p=0.0022 for 6c and 6d respectively. For reference, the students t-test had slightly worse statistics p=0.040 and p=0.0023. The score remained the same 1 and 2 stars respectively.

The references provided for the statement regarding cascade activation of the TRPs are incredibly out of date. While it is clear that TRPV4 can be activated by a second messenger cascade downstream of osmotic swelling of cells, TRPV4 has also been shown to be activated by mechanical inputs at the cell-substrate interface, even when the second messenger cascade is inhibited. Recommend updating the references to reflect more current understanding of channel activation.

Response: We thank the reviewer for pointing this out. We have updated the references and changed the comment to “can be” instead of “are”. The reference is more general to multiple ion channel types including KCNQ4. This should avoid any perceived conflict with the cellsubstrate interface mechanism which we very much agree is a correct mechanism for TRP channels.

Minor comments re text editing etc:The central messages of the manuscript would benefit from extensive work to increase the precision of the writing of the manuscript and the presentation of data in the figures, such textual changes alone would help address a number of the concerns outlined in this review, by clarifying some ambiguities. There are numerous errors throughout, ranging from grammatical issues, ambiguities with definitions, lack of scale bars in images, lack of labels on graph axes, lack of clarity due to the mode of presentation of sample numbers (it would be far more precise to indicate specific numbers for each sample rather than a range, which is ambiguous and confusing), unnecessary and repeat information in the methods section. Below are some examples but this list is not exhaustive.

Response: Thank you, reviewer # 1 also had many of these concerns. We have gone through the entire paper and improved the precision of the writing of the manuscript. We have also added the missing error bar to Figure 6. And axis labels have been added to the inset images. The redundancy in cell culture methods has been removed. Where a range is small and there are lots of values, the exact number of ‘n’ are graphically displayed in the dot plot for each condition.

Text:I recommend considering how to discuss the various aspects of channel activation. A convention in the field is to use mechanical activation or mechanical gating to describe that process where the mechanical stimulus is directly coupled to the channel gating mechanism. This would be the case for the activation of TREK-1 by membrane stretch alone. The increase in activation by PLD2 activity then reflects a modulation of the mechanical activation of the channel, because the relevant gating stimulus is PA, rather than force/stretch. The sum of these events could be described as shear-evoked or mechanically-evoked, TREK-1 mediated currents (thus making it clear that the mechanical stimulus initiates the relevant cascade, but the gating stimulus may be other than direct mechanical input.) Given the interesting and compelling data offered in this manuscript regarding the sensitisation of TREK-1 dependent mechanicallyevoked currents by PLD2, an increase in the precision of the language would help convey the central message of this work.

Response; We agree there needs to be convention. We have taken the suggestion of mechanically evoked and we suggest the following definitions:

1. Mechanical activation of PLD2: direct force on the lipids releasing PLD2 from nonactivating lipids.

2. Mechanical activation/gating of TREK1: direct force from lipids from either tension or hydrophobic mismatch that opens the channel.

3. Mechanically evoked: a mechanical event that leads to a downstream effect. The effect is mechanically “evoked”.

4. Spatial patterning/biochemistry: nanoscopic changes in the association of a protein with a nanoscopic lipid cluster or compartment.

An example of where discussion of mechanical activation is ambiguous in the text is found at line 109: "channel could be mechanically activated by a movement from GM1 to PIP2 lipids." In this case, the sentence could be suggesting that the movement between lipids provides the mechanical input that activates the channel, which is not what the data suggest.

Response: Were possible we have replaced “movement” with “spatial patterning” and “association” and “dissociation” from specific lipid compartment. This better reflects the data we have in this paper. However, we do think that a movement mechanically activates the channel, GM1 lipids are thick and PIP2 lipids are thin, so movement between the lipids could activate the channel through direct lipid interaction. We will address this aspect in a future paper.

Inconsistencies with usage:• TREK1 versus TREK-1

Response: corrected to TREK-1

• mPLD2 versus PLD2

Response: where PLD2 represents mouse this has been corrected.

• K758R versus xPLD2

Response: we replaced K758R in the methods with xPLD2.

• HEK293T versus HEK293t Response: we have changed all instances to read HEK293T.• Drosophila melanogaster and *D. melanogaster* used inconsistently and in many places incorrectly

Response: we have read all to read the common name *Drosophila*.

Line 173: misspelled methylbetacyclodextrin

Response corrected

Line 174: degree symbol missing

Response corrected

Line 287: "the decrease in cholesterol likely evolved to further decrease the palmate order in the palmitate binding site"... no evidence, no support for this statement, falsely attributes intention to evolutionary processes .

Response: we have removed the reference to evolution at the request of the reviewer, it is not necessary. But we do wish to note that to our knowledge, all biological function is scientifically attributed to evolution. The fact that cholesterol decreases in response to shear is evidence alone that the cell evolved to do it.

Line 307: grammatical error

Response: the redundant Lipid removed.

Line 319: overinterpreted - how is the mechanosensitivy of GPCRs explained by this translocation?

Response: all G-alpha subunits of the GPCR complex are palmitoylated. We showed PLD (which has the same lipidation) is mechanically activated. If the palmitate site is disrupted for PLD2, then it is likely disrupted for every G-alpha subunit as well.

Line 582: what is the wild type referred to here?

Response: human full length with a GFP tag.

Methods:• Sincere apologies if I missed something but I do not recall seeing any experiments using purified TREK-1 or flux assays. These details should be removed from the methods section

Response: Removed.

• There is significant duplication of detail across the methods (three separate instances of electrophysiology details) these could definitely be consolidated.

Response: Duplicates removed.

Figures:• Figure 2- b box doesn't correspond to inset. Bottom panel should provide overview image for the cell that was assessed with shear. In bottom panel, circle outlines an empty space.

Response: We have widened the box slightly to correspond so the non shear box corresponds to the middle panel. We have also added the picture for the whole cell to Fig S2g and outlined the zoom shown in the bottom panel of Fig 2b as requested. The figure is of the top of a cell. We also added the whole cell image of a second sheared cell.

**Author response image 1. sa4fig1:** 

• Figure 3 b+c: inset graph lacking axis labels

Response; the inset y axis is the same as the main axis. We added “pair corr. (5nM)” and a description in the figure legend to make this clearer. The purpose of the inset is to show statistical significance at a single point. The contrast has been maximized but without zooming in points can be difficult to see.

• Figure 5: replicate numbers missing and individual data points lacking in panels b + c, no labels of curve in b + c, insets, unclear what (5 nm) refers to in insets.

Response: Thank you for pointing out these errors. The N values have been added. Similar to figure 3, the inset is a bar graph of the pair correlation data at 5 nm. A better explanation of the data has been added to the figure legend.

• Figure 6: no scale bar, no clear membrane localization evident from images presented, panel g offers virtually nothing in terms of insight

Response: We have added scale bars to figure 6b. Figure 6g is intentionally simplistic, we found that correlating decreased threshold with increased pain was confusing. A previous reviewer claimed our data was inconsistent. The graphic avoids this confusion. We also added negative effects of low PA on downstream effects to the right panel. This helps graphically show we don’t know the downstream effects.

**Reviewer #3 (Recommendations For The Authors):**
Minor suggestions:1. line 162, change 'heat' to 'temperature'.

Response: changed.

1. in figure 1, it would be helpful to keep the unit for current density consistent among different panels. 1e is a bit confusing: isn't the point of Figure 1 that most of TREK1 activation is not caused by direct force-sensing?

Response: Yes, the point of figure 1 is to show that in a biological membrane over expressed TREK-1 is a downstream effector of PLD2 mechanosensation which is indirect. We agree the figure legend in the previous version of the paper is very confusing.

There is almost no PLD2 independent current in our over expressed system, which is represented by no ions in the conduction pathway of the channel despite there being tension on the membrane.

Purified TREK-1 is only mechanosensitive in a few select lipids, primarily crude Soy PC. It was always assumed that HEK293 and Cos cells had the correct lipids since over expressed TREK-1 responded to mechanical force in these lipids. But that does not appear to be correct, or at least only a small amount of TREK-1 is in the mechanosensitive lipids. Figure 1e graphically shows this. The arrows indicate tension, but the channel isn’t open with xPLD2 present. We added a few sentences to the discussion to further clarify.

Panels c has different units because the area of the tip was measured whereas in d the resistance of the tip was measured. They are different ways for normalizing for small differences in tip size.

1. line 178, ~45 of what?

Response: Cells were fixed for ~30 sec.

1. line 219 should be Figure 4f?

Response: thank you, yes Figure 4f.

Previous public reviews with minor updates.

**Reviewer #1 (Public Review):**
Force sensing and gating mechanisms of the mechanically activated ion channels is an area of broad interest in the field of mechanotransduction. These channels perform important biological functions by converting mechanical force into electrical signals. To understand their underlying physiological processes, it is important to determine gating mechanisms, especially those mediated by lipids. The authors in this manuscript describe a mechanism for mechanically induced activation of TREK-1 (TWIK-related K+ channel). They propose that force induced disruption of ganglioside (GM1) and cholesterol causes relocation of TREK-1 associated with phospholipase D2 (PLD2) to 4,5-bisphosphate (PIP2) clusters, where PLD2 catalytic activity produces phosphatidic acid that can activate the channel. To test their hypothesis, they use dSTORM to measure TREK-1 and PLD2 colocalization with either GM1 or PIP2. They find that shear stress decreases TREK-1/PLD2 colocalization with GM1 and relocates to cluster with PIP2. These movements are affected by TREK-1 C-terminal or PLD2 mutations suggesting that the interaction is important for channel re-location. The authors then draw a correlation to cholesterol suggesting that TREK-1 movement is cholesterol dependent. It is important to note that this is not the only method of channel activation and that one not involving PLD2 also exists. Overall, the authors conclude that force is sensed by ordered lipids and PLD2 associates with TREK-1 to selectively gate the channel. Although the proposed mechanism is solid, some concerns remain.1. Most conclusions in the paper heavily depend on the dSTORM data. But the images provided lack resolution. This makes it difficult for the readers to assess the representative images.

Response: The images were provided are at 300 dpi. Perhaps the reviewer is referring to contrast in Figure 2? We are happy to increase the contrast or resolution.

As a side note, we feel the main conclusion of the paper, mechanical activation of TREK-1 through PLD2, depended primarily on the electrophysiology in Figure 1b-c, not the dSTORM. But both complement each other.

1. The experiments in Figure 6 are a bit puzzling. The entire premise of the paper is to establish gating mechanism of TREK-1 mediated by PLD2; however, the motivation behind using flies, which do not express TREK-1 is puzzling.

Response: The fly experiment shows that PLD mechanosensitivity is more evolutionarily conserved than TREK-1 mechanosensitivity. We have added this observation to the paper.

-Figure 6B, the image is too blown out and looks over saturated. Unclear whether the resolution in subcellular localization is obvious or not.

Response: Figure 6B is a confocal image, it is not dSTORM. There is no dSTORM in Figure 6. We have added the error bars to make this more obvious. For reference, only a few cells would fit in the field of view with dSTORM.

-Figure 6C-D, the differences in activity threshold is 1 or less than 1g. Is this physiologically relevant? How does this compare to other conditions in flies that can affect mechanosensitivity, for example?

Response: Yes, 1g is physiologically relevant. It is almost the force needed to wake a fly from sleep (1.2-3.2g). See ref 33. Murphy Nature Pro. 2017.

1. 70mOsm is a high degree of osmotic stress. How confident are the authors that a cell health is maintained under this condition and b. this does indeed induce membrane stretch? For example, does this stimulation activate TREK-1?

Response: Yes, osmotic swell activates TREK1. This was shown in ref 19 (Patel et al 1998). We agree the 70 mOsm is a high degree of stress. This needs to be stated better in the paper.

**Reviewer #2 (Public Review):**
This manuscript by Petersen and colleagues investigates the mechanistic underpinnings of activation of the ion channel TREK-1 by mechanical inputs (fluid shear or membrane stretch) applied to cells. Using a combination of super-resolution microticopy, pair correlation analysis and electrophysiology, the authors show that the application of shear to a cell can lead to changes in the distribution of TREK-1 and the enzyme PhospholipaseD2 (PLD2), relative to lipid domains defined by either GM1 or PIP2. The activation of TREK-1 by mechanical stimuli was shown to be sensi>zed by the presence of PLD2, but not a catalytically dead xPLD2 mutant. In addition, the activity of PLD2 is increased when the molecule is more associated with PIP2, rather than GM1 defined lipid domains. The presented data do not exclude direct mechanical activation of TREK-1, rather suggest a modulation of TREK-1 activity, increasing sensitivity to mechanical inputs, through an inherent mechanosensitivity of PLD2 activity. The authors additionally claim that PLD2 can regulate transduction thresholds in vivo using *Drosophila melanogaster* behavioural assays. However, this section of the manuscript overstates the experimental findings, given that it is unclear how the disruption of PLD2 is leading to behavioural changes, given the lack of a TREK-1 homologue in this organism and the lack of supporting data on molecular function in the relevant cells.

Response: We agree, the downstream effectors of PLD2 mechanosensitivity are not known in the fly. Other anionic lipids have been shown to mediate pain see ref 46 and 47. We do not wish to make any claim beyond PLD2 being an in vivo contributor to a fly’s response to mechanical force. We have removed the speculative conclusions about fly thresholds from the abstract.

That said we do believe we have established a molecular function at the cellular level. We showed PLD is robustly mechanically activated in a cultured fly cell line (BG2-c2) Figure 6a of the manuscript. And our previous publication established mechanosensation of PLD (Petersen et. al. Nature Com 2016) through mechanical disruption of the lipids. At a minimum, the experiments show PLDs mechanosensitivity is evolutionarily better conserved across species than TREK1.

This work will be of interest to the growing community of scientists investigating the myriad mechanisms that can tune mechanical sensitivity of cells, providing valuable insight into the role of functional PLD2 in sensi>zing TREK-1 activation in response to mechanical inputs, in some cellular systems.The authors convincingly demonstrate that, post application of shear, an alteration in the distribution of TREK-1 and mPLD2 (in HEK293T cells) from being correlated with GM1 defined domains (no shear) to increased correlation with PIP2 defined membrane domains (post shear). These data were generated using super-resolution microticopy to visualise, at sub diffraction resolution, the localisation of labelled protein, compared to labelled lipids. The use of super-resolution imaging enabled the authors to visualise changes in cluster association that would not have been achievable with diffraction limited microticopy. However, the conclusion that this change in association reflects TREK-1 leaving one cluster and moving to another overinterprets these data, as the data were generated from sta>c measurements of fixed cells, rather than dynamic measurements capturing molecular movements.When assessing molecular distribution of endogenous TREK-1 and PLD2, these molecules are described as "well correlated: in C2C12 cells" however it is challenging to assess what "well correlated" means, precisely in this context. This limitation is compounded by the conclusion that TREK-1 displayed little pair correlation with GM1 and the authors describe a "small amount of TREK-1 trafficked to PIP2". As such, these data may suggest that the findings outlined for HEK293T cells may be influenced by artefacts arising from overexpression.The changes in TREK-1 sensitivity to mechanical activation could also reflect changes in the amount of TREK-1 in the plasma membrane. The authors suggest that the presence of a leak currently accounts for the presence of TREK-1 in the plasma membrane, however they do not account for whether there are significant changes in the membrane localisation of the channel in the presence of mPLD2 versus xPLD2. The supplementary data provide some images of fluorescently labelled TREK-1 in cells, and the authors state that truncating the c-terminus has no effect on expression at the plasma membrane, however these data provide inadequate support for this conclusion. In addition, the data reporting the P50 should be noted with caution, given the lack of saturation of the current in response to the stimulus range.

Response: We thank the reviewer for his/her concern about expression levels. We did test TREK-1 expression. mPLD decreases TREK-1 expression ~two-fold (see Author response image 2 below). We did not include the mPLD data since TREK-1 was mechanically activated with mPLD. For expression to account for the loss of TREK-1 stretch current (Figure 1b), xPLD would need to block surface expression of TREK-1 prior to stretch. The opposite was true, xPLD2 increased TREK-1 expression (see Figure S2c). Furthermore, we tested the leak current of TREK-1 at 0 mV and 0 mmHg of stretch. Basal leak current was no different with xPLD2 compared to endogenous PLD (Figure 1d; red vs grey bars respectively) suggesting TREK-1 is in the membrane and active when xPLD2 is present. If anything, the magnitude of the effect with xPLD would be larger if the expression levels were equal.

**Author response image 2. sa4fig2:** TREK expression at the plasma membrane. TREK-1 Fluorescence was measured by GFP at points along the plasma membrane. Over expression of mouse PLD2 (mPLD) decrease the amount of full-length TREK-1 (FL TREK) on the surface more than 2-fold compared to endogenously expressed PLD (enPLD) or truncated TREK (TREKtrunc) which is missing the PLD binding site in the C-terminus. Over expression of mPLD had no effect on TREKtrunc.

Finally, by manipulating PLD2 in *D. melanogaster*, the authors show changes in behaviour when larvae are exposed to either mechanical or electrical inputs. The depletion of PLD2 is concluded to lead to a reduction in activation thresholds and to suggest an in vivo role for PA lipid signaling in setting thresholds for both mechanosensitivity and pain. However, while the data provided demonstrate convincing changes in behaviour and these changes could be explained by changes in transduction thresholds, these data only provide weak support for this specific conclusion. As the authors note, there is no TREK-1 in *D. melanogaster*, as such the reported findings could be accounted for by other explanations, not least including potential alterations in the activation threshold of Nav channels required for action potential generation. To conclude that the outcomes were in fact mediated by changes in mechanotransduction, the authors would need to demonstrate changes in receptor potential generation, rather than deriving conclusions from changes in behaviour that could arise from alterations in resting membrane potential, receptor potential generation or the activity of the voltage gated channels required for action potential generation.

Response: We are willing to restrict the conclusion about the fly behavior as the reviewers see fit. We have shown PLD is mechanosensitivity in a fly cell line, and when we knock out PLD from a fly, the animal exhibits a mechanosensation phenotype. We tried to make it clear in the figure and in the text that we have no evidence of a particular mechanism downstream of PLD mechanosensation.

This work provides further evidence of the astounding flexibility of mechanical sensing in cells. By outlining how mechanical activation of TREK-1 can be sensitised by mechanical regulation of PLD2 activity, the authors highlight a mechanism by which TREK-1 sensitivity could be regulated under distinct physiological conditions.
**Reviewer #3 (Public Review):**
The manuscript "Mechanical activation of TWIK-related potassium channel by nanoscopic movement and second messenger signaling" presents a new mechanism for the activation of TREK-1 channel. The mechanism suggests that TREK1 is activated by phosphatidic acids that are produced via a mechanosensitive motion of PLD2 to PIP2-enriched domains. Overall, I found the topic interesting, but several typos and unclarities reduced the readability of the manuscript. Additionally, I have several major concerns on the interpretation of the results. Therefore, the proposed mechanism is not fully supported by the presented data. Lastly, the mechanism is based on several previous studies from the Hansen lab, however, the novelty of the current manuscript is not clearly stated. For example, in the 2nd result section, the authors stated, "fluid shear causes PLD2 to move from cholesterol dependent GM1 clusters to PIP2 clusters and this activated the enzyme". However, this is also presented as a new finding in section 3 "Mechanism of PLD2 activation by shear."For PLD2 dependent TREK-1 activation. Overall, I found the results compelling.However, two key results are missing.1. Does HEK cells have endogenous PLD2? If so, it's hard to claim that the authors can measure PLD2-independent TREK1 activation.

Response: yes, there is endogenous PLD (enPLD). We calculated the relative expression of xPLD2 vs enPLD. xPLD2 is >10x more abundant (Fig. S3d of Pavel et al PNAS 2020, ref 14 of the current manuscript). Hence, as with anesthetic sensitivity, we expect the xPLD to out compete the endogenous PLD, which is what we see. We added the following sentence and reference : “The xPLD2 expression is >10x the endogenous PLD2 (enPLD2) and out computes the TREK-1 binding site for PLD25.”

1. Does the plasma membrane trafficking of TREK1 remain the same under different conditions (PLD2 overexpression, truncation)? From Figure S2, the truncated TREK1 seem to have very poor trafficking. The change of trafficking could significantly contribute to the interpretation of the data in Figure 1.

Response: If the PLD2 binding site is removed (TREK-1trunc), yes, the trafficking to the plasma membrane is unaffected by the expression of xPLD and mPLD (Author response image 2 above). For full length TREK1 (FL-TREK-1), co-expression of mPLD decreases TREK expression (Author response image 2) and coexpression with xPLD increases TREK expression (Figure S2f). This is exactly opposite of what one would expect if surface expression accounted for the change in pressure currents. Hence, we conclude surface expression does not account for loss of TREK-1 mechanosensitivity with xPLD2. A few sentences was added to the discussion. We also performed dSTORM on the TREKtruncated using EGFP. TREK-truncated goes to PIP2 (see figure 2 of 6)

**Author response image 3. sa4fig3:** 

To better compare the levels of TREK-1 before and after shear, we added a supplemental figure S2f where the protein was compared simultaneously in all conditions. 15 min of shear significantly decreased TREK-1 except with mPLD2 where the levels before shear were already lowest of all the expression levels tested.

For shear-induced movement of TREK1 between nanodomains. The section is convincing, however I'm not an expert on super-resolution imaging. Also, it would be helpful to clarify whether the shear stress was maintained during fixation. If not, what is the >me gap between reduced shear and the fixed state. lastly, it's unclear why shear flow changes the level of TREK1 and PIP2.

Response: Shear was maintained during the fixing. xPLD2 blocks endocytosis, presumably endocytosis and or release of other lipid modifying enzymes affect the system. The change in TREK-1 levels appears to be directly through an interaction with PLD as TREK trunc is not affected by over expression of xPLD or mPLD.

For the mechanism of PLD2 activation by shear. I found this section not convincing. Therefore, the question of how does PLD2 sense mechanical force on the membrane is not fully addressed. Par>cularly, it's hard to imagine an acute 25% decrease cholesterol level by shear - where did the cholesterol go? Details on the measurements of free cholesterol level is unclear and additional/alternative experiments are needed to prove the reduction in cholesterol by shear.

Response: The question “how does PLD2 sense mechanical force on the membrane” we addressed and published in Nature Comm. In 2016. The title of that paper is “Kinetic disruption of lipid rafts is a mechanosensor for phospholipase D” see ref 13 Petersen et. al. PLD is a soluble protein associated to the membrane through palmitoylation. There is no transmembrane domain, which narrows the possible mechanism of its mechanosensation to disruption.

The Nature Comm. reviewer identified as “an expert in PLD signaling” wrote the following of our data and the proposed mechanism:

“This is a provocative report that identi0ies several unique properties of phospholipase D2 (PLD2). It explains in a novel way some long established observations including that the enzyme is largely regulated by substrate presentation which 0its nicely with the authors model of segregation of the two lipid raft domains (cholesterol ordered vs PIP2 containing). Although PLD has previously been reported to be involved in mechanosensory transduction processes (as cited by the authors) this is the 0irst such report associating the enzyme with this type of signaling... It presents a novel model that is internally consistent with previous literature as well as the data shown in this manuscript. It suggests a new role for PLD2 as a force transduction tied to the physical structure of lipid rafts and uses parallel methods of disrup0on to test the predic0ons of their model.”

Regarding cholesterol. We use a fluorescent cholesterol oxidase assay which we described in the methods. This is an appropriate assay for determining cholesterol levels in a cell which we use routinely. We have published in multiple journals using this method, see references 28, 30, 31. Working out the metabolic fate of cholesterol after sheer is indeed interesting but well beyond the scope of this paper. Furthermore, we indirectly confirmed our finding using dSTORM cluster analysis (Figure 3d-e). The cluster analysis shows a decrease in GM1 cluster size consistent with our previous experiments where we chemically depleted cholesterol and saw a similar decrease in cluster size (see ref 13). All the data are internally consistent, and the cholesterol assay is properly done. We see no reason to reject the data.

Importantly, there is no direct evidence for "shear thinning" of the membrane and the authors should avoid claiming shear thinning in the abstract and summary of the manuscript.

Response: We previously established a kinetic model for PLD2 activation see ref 13 (Petersen et al Nature Comm 2016). In that publication we discussed both entropy and heat as mechanisms of disruption. Here we controlled for heat which narrowed that model to entropy (i.e., shear thinning) (see Figure 3c). We provide an overall justification below. But this is a small refinement of our previous paper, and we prefer not to complicate the current paper. We believe the proper rheological term is shear thinning. The following justification, which is largely adapted from ref 13, could be added to the supplement if the reviewer wishes.

Justification: To establish shear thinning in a biological membrane, we initially used a soluble enzyme that has no transmembrane domain, phospholipase D2 (PLD2). PLD2 is a soluble enzyme and associated with the membrane by palmitate, a saturated 16 carbon lipid attached to the enzyme. In the absence of a transmembrane domain, mechanisms of mechanosensation involving hydrophobic mismatch, tension, midplane bending, and curvature can largely be excluded. Rather the mechanism appears to be a change in fluidity (i.e., kinetic in nature). GM1 domains are ordered, and the palmate forms van der Waals bonds with the GM1 lipids. The bonds must be broken for PLD to no longer associate with GM1 lipids. We established this in our 2016 paper, ref 13. In that paper we called it a kinetic effect, however we did not experimentally distinguish enthalpy (heat) vs. entropy (order). Heat is Newtonian and entropy (i.e., shear thinning) is non-Newtonian. In the current study we paid closer attention to the heat and ruled it out (see Figure 3c and methods). We could propose a mechanism based on kinetic disruption, but we know the disruption is not due to melting of the lipids (enthalpy), which leaves shear thinning (entropy) as the plausible mechanism.

The authors should also be aware that hypotonic shock is a very dirty assay for stretching the cell membrane. Ouen, there is only a transient increase in membrane tension, accompanied by many biochemical changes in the cells (including acidification, changes of concentration etc). Therefore, I would not consider this as definitive proof that PLD2 can be activated by stretching membrane.

Response: Comment noted. We trust the reviewer is correct. In 1998 osmotic shock was used to activate the channel. We only intended to show that the system is consistent with previous electrophysiologic experiments.

References cited:

1 Du G, Huang P, Liang BT, Frohman MA. Phospholipase D2 localizes to the plasma membrane and regulates angiotensin II receptor endocytosis. Mol Biol Cell 2004;15:1024–30. https://doi.org/10.1091/mbc.E03-09-0673.

2 Koch T, Wu DF, Yang LQ, Brandenburg LO, Höllt V. Role of phospholipase D2 in the agonist-induced and constistutive endocytosis of G-protein coupled receptors. J Neurochem 2006;97:365–72. https://doi.org/10.1111/j.1471-4159.2006.03736.x.

3 Wheeler DS, Underhill SM, Stolz DB, Murdoch GH, Thiels E, Romero G, et al. Amphetamine activates Rho GTPase signaling to mediate dopamine transporter internalization and acute behavioral effects of amphetamine. Proc Natl Acad Sci U S A 2015;112:E7138–47. https://doi.org/10.1073/pnas.1511670112.

4 Rankovic M, Jacob L, Rankovic V, Brandenburg L-OO, Schröder H, Höllt V, et al. ADP-ribosylation factor 6 regulates mu-opioid receptor trafficking and signaling via activation of phospholipase D2. Cell Signal 2009;21:1784–93. https://doi.org/10.1016/j.cellsig.2009.07.014.

5 Pavel MA, Petersen EN, Wang H, Lerner RA, Hansen SB. Studies on the mechanism of general anesthesia. Proc Natl Acad Sci U S A 2020;117:13757–66. https://doi.org/10.1073/pnas.2004259117.

6 Call IM, Bois JL, Hansen SB. Super-resolution imaging of potassium channels with genetically encoded EGFP. BioRxiv 2023. https://doi.org/10.1101/2023.10.13.561998.